



# [10]Be systematics in the Tsangpo-Brahmaputra catchment: the cosmogenic nuclide legacy of the eastern Himalayan syntaxis

Maarten Lupker[1,6], Jérôme Lavé[2], Christian France-Lanord[2], Marcus Christl[3], Didier Bourlès[4], Julien Carcaillet[5], Colin Maden[6], Rainer Wieler[6], Mustafizur Rahman[7], Devojit Bezbaruah[8], Liu Xiaohan[9]

[1]Geological Institute, D-ERDW, ETH Zürich, Zürich, 8092, Switzerland
[2]CRPG, UMR 7358 CNRS-Univ. de Lorraine, Vandoeuvre les Nancy, 54500, France
[3]Institute of Particle Physics, D-PHYS, ETH Zürich, Zürich, 8093, Switzerland
[4]CEREGE, UMR 34 UAM-CNRS-IRD, Aix-en-Provence, 13545, France
[5]ISTerre, Univ. Grenoble Alpes-CNRS, Grenoble, 38000, France
[6]Insitute of Geochemistry and Petrology, D-ERDW, ETH Zürich, Zürich, 8092, Switzerland
[7]Department of Soil, Water and Environment, Dhaka University, Dhaka, 1000, Bangladesh
[8]Department of Applied Geology, Dibrugarh University, Dibrugarh, 786004, India
[9]Institute of Tibetan Plateau Research, Chinese Academy of Sciences, Beijing, China

*Correspondence to*: Maarten Lupker (maarten.lupker@erdw.ethz.ch)

**Abstract.** The Tsangpo-Brahmaputra River drains the eastern part of the Himalayan range, and flows from the Tibetan Plateau through the eastern Himalayan syntaxis downstream to the Indo-Gangetic floodplain and the Bay of Bengal. As such it is a unique natural laboratory to study how denudation and sediment production processes are transferred to river detrital signals. In this study, we present a new [10]Be data set to constrain denudation rates across the catchment and to quantify the impact of rapid erosion within the syntaxis region on cosmogenic nuclide budgets and signals. [10]Be denudation rates span

around two orders of magnitude across the catchments (ranging from 0.03 mm/yr to > 4 mm/yr) and sharply increase as the Tsangpo-Brahmaputra flows across the eastern Himalaya. The increase in denudation rates however occurs ~150 km downstream of the Namche Barwa-Gyala Peri massif (NBGPm), an area which has been previously characterized by extremely high erosion and exhumation rates. We suggest that this downstream lag is mainly due to the physical abrasion of coarse grained, low [10]Be concentration, landslide material produced within the syntaxis that dilutes the upstream high

concentration [10]Be flux from the Tibetan Plateau only after abrasion has transferred sediment to the studied sand fraction. A simple abrasion model produces typical lag distances of 50 to 150 km compatible with our observations. Abrasion effects reduce the spatial resolution over which denudation can be constrained in the eastern Himalayan syntaxis. In addition, we also highlight that denudation rate estimates are dependent on the sediment connectivity, storage and quartz content of the upstream Tibetan Plateau part of the catchment which tends to lead to an overestimation of downstream denudations rates.

Taking these effects into account we estimate a denudation rates of ca. 2 to 5 mm/yr for the entire syntaxis and ca. 4 to 28 mm/yr for the NBGPm, which is significantly higher than other to other large catchments. Overall, [10]Be concentrations measured at the outlet of the Tsangpo-Brahmaputra in Bangladesh suggest a sediment flux between 780 and 1430 Mt/yr equivalent to a denudation rate between 0.7 and 1.2 mm/yr for the entire catchment.

## 1 Introduction

The Himalayan range is the largest point source of sediments to the ocean as a consequence of intense tectonic and climatic forcing, sustaining high erosion rates and sediment production. The Tsangpo-Brahmaputra is a major catchment draining the Himalayan range and the Tibetan Plateau. It is an exceptional natural laboratory to better understand how denudation processes are transferred to and reflected in the sedimentary load because of its very contrasted catchment area in terms of physiography, geomorphological processes and climate. A remarkable feature of the Tsangpo-Brahmaputra catchment is the

eastern Himalayan syntaxis. Mountain belt syntaxes are regions where the interactions between tectonic and surface processes are most likely to be apparent (Koons et al., 2013). The eastern Himalayan syntaxis has been proposed as a typical





example where active tectonic deformation, thermal weakening of the crust and steep topography could be self-sustained by intense erosion and rapid exhumation of crustal material within the framework of the tectonic aneurysm model (Zeitler et al., 2001). Although the extent and nature of this coupling has recently been challenged (Bendick and Ehlers, 2014; Wang et al., 2014; King et al., 2016), there is ample evidence for superimposed rapid exhumation (Burg et al., 1998; Seward & Burg, 2008; Zeitler et al., 2014; Bracciali et al., 2016) and active erosion (Finlayson et al., 2002; Finnegan et al., 2008; Stewart et al., 2008; Enkelmann et al., 2011; Larsen and Montgomery, 2012; Lang et al., 2013) in a focused area around the Namche Barwa-Gyala Peri massif (NBGPm), where the course of the Tsangpo-Brahmaputra is marked by sharp bend to the south west (Fig. 1). Difficult access to the NBGPm means that contemporary denudation rates are poorly constrained but estimates range from ca. 5 to 17 mm/yr (Stewart et al., 2008; Enkelmann et al., 2011; Larsen and Montgomery, 2012). The NBGPm is thought to be responsible for up to 40 to 70 % of the sediment discharge of the entire Brahmaputra (e.g. Singh and France-Lanord, 2002; Garzanti et al., 2004) illustrating the potential for a small area of the Tsangpo-Brahmaputra to act as a major source of sediments to the entire river network. Upstream of the NBGPm, the Tsangpo-Brahmaputra drains the relatively flat, slowly eroding and arid southern edge of the Tibet Plateau, while the downstream part of the catchments includes a range of large Himalayan tributaries draining the eastern half of the Himalayan range and the Indo-Gangetic floodplain. In this work, we present a terrestrial cosmogenic nuclide (TCN), $^{10}$Be, data-set spanning the entire Tsangpo-Brahmaputra catchment, covering the Tibetan Plateau downstream to the outlet in Bangladesh. The wide range of surface processes and rates of landscape evolution across the catchment represents a unique opportunity to study how these denudation signals are transferred through the catchment and integrated into the sedimentary load. The use of TCN also represents an opportunity to provide new quantitative constraints on the denudation rates across the catchment in general and the NBGPm in particular.

Cosmogenic nuclides have been widely used to constrain catchment wide denudation rates (CWDR) over a range of scales and catchment sizes (e.g. Portenga et al., 2011). The concentration of a given cosmogenic nuclide in river sediments theoretically reflects the rate at which the upstream landscape is lowering through physical erosion and chemical weathering assuming steady state denudation (Brown et al., 1995; Granger et al., 1996). However, it is also increasingly recognized that CWDR are affected by a number of site-specific biases and limitations that prevents us from "simply reading denudation rates from a bag of sand". Amongst others, landsliding, a dominant erosion process in actively eroding landscapes, violates the steady state denudation assumption and may thus lead to biases in the CWDR (Niemi et al., 2005; Yanites et al., 2009). The stochastic nature of sediment supply to the channels and hydrology within the river network may also perturb the TCN signal over a range of time scales and limits our ability to use a single sediment sample as a representative and accurate representative of upstream erosion products (Kober et al., 2012; Lupker et al., 2012; West et al., 2014; Foster and Anderson, 2016). Glacial erosion is not properly accounted for, as sediments eroded under a glacier are shielded from cosmic rays and hence their TCN concentration does not relate to denudation which affects the TCN budget further downstream (Godard et al., 2012; Delunel et al., 2014). Different erosion processes also affect the final grain-size of fluvially exported sediment, which may in turn lead to a grain-size dependence of TCN signals that needs to be taken into account in order to derive robust CWDR (Clapp et al., 2002; Aguilar et al., 2014; Puchol et al., 2014). The assumption of quartz ubiquity or content within the eroded lithology of a studied catchment is also likely to be violated in a number of cases and is also a source of biases in CWDR (Carretier et al., 2015). Finally, large catchments and especially floodplains represent temporary storage compartments, where TCN may accumulate or decay depending on storage duration and depth, which may in turn affect the TCN concentration in river sediments (Wittmann and von Blanckenburg, 2016). These limitations need to be properly tested and accounted for in order to explore the full potential of TCN as quantitative tracers of surface processes and denudation rates. Since most of these limitations also apply to the Tsangpo-Brahmaputra, this study further represents a unique opportunity to evaluate and quantify the potential biases associated with the use of TCNs in large, actively eroding and contrasted catchments.



## 2 Settings and methods

### 2.1 The Tsangpo-Brahmaputra catchment

The Yarlung – Tsangpo – Siang – Brahmaputra – Jamuna River system (as the trunk river is called from up- to downstream,

further referred to as the Tsangpo-Brahmaputra in this manuscript) is a large drainage basin of ca. 530000 km², draining the eastern half of the Himalayan range. It originates close to Mount Kailash, in central Tibet, and flows eastward along the Indus-Tsangpo Suture zone and the northern flank of the Himalayan range for about 1500 km. This part of the catchment is marked by a relatively flat topography and a high average elevation (> 4000 m a.s.l.). Being in the rain shadow of the Himalayan range, it receives little precipitation (< 500 mm/yr) and a significant part of the runoff is generated by snowmelt

from the range to the north (Bookhagen and Burbank, 2010). The Tibetan part of the Tsangpo-Brahmaputra catchment is dominated by Paleozoic sedimentary sequences with the carbonate-rich Tethyan Sedimentary Series (TSS) and ultramafic to felsic rocks of the Trans Himalayan batholiths and the Neotethys ophiolites (Liu and Einsele, 1994).

In the north-eastern part of the catchment, the Tsangpo-Brahmaputra River sharply bends to the south, leaves the Tibetan

Plateau and cuts through the Himalayan range through deep gorges and a prominent knickpoint, losing over 2000 m of elevation in less than 100 km of channel length. The NBGPm, and more generally the eastern Himalayan syntaxis, is marked by very steep topography and high relief resulting from the deflection of crustal material around the indenting Indian plate and the growth of a large antiformal metamorphic structure that is deeply dissected by active fluvial incision (Burg et al., 1997; Finnegan et al., 2008). This reach of the Tsangpo-Brahmaputra is underlain by gneiss, quartzites, marbles and

generally highly metamorphic rocks from the Higher Himalayan Crystallines (HHC) (Burg et al., 1997). Just south of the syntaxis, the river flows over the Abor volcanics as well as limestones and shales (Bhat, 1984). The large valley along the Tsangpo-Brahmaputra, south of the syntaxis, channelizes moisture and precipitation northwards, which translates into significant rainfall (> 2000 mm/yr) up to 100 km north of the Himalayan front in the syntax region (Anders et al., 2006; Bookhagen et al., 2006).

The Tsangpo-Brahmaputra exits the Himalayan range in Pasighat, NE India, and finally discharges into the Bay of Bengal in Bangladesh after crossing the Assam and Indo-Gangetic alluvial floodplains for about 1000 km (Fig. 1). The main stream Tsangpo-Brahmaputra further receives input from the eastern Himalayan Rivers Lohit and Dibang that drain the poorly documented Mishmi hill formations as well as input from Himalayan rivers draining the southern flank of the eastern

Himalayan range, such as the Subansiri, Kameng, Manas and Teesta Rivers. The northern tributaries of the Tsangpo-Brahmaputra drain the classical HHC and Lesser Himalaya units composed of a variety of crystalline and metasedimentary rocks, as well as the ca. mid Miocene sub-Himalayan Siwalik molasses (Yin et al., 2006; Chirouze et al., 2012). Sediment input from the southern tributaries originating from the Shillong Plateau, as well as from the northern part of the Indo-Burman ranges are thought to be small (Garzanti et al., 2004). The Indo-Gangetic floodplain and the southern flank

tributaries to the Tsangpo-Brahmaputra receive intense precipitation (up to 4000 mm/yr), which mainly falls during the 4 months of the Indian summer monsoon (June to September) (Bookhagen and Burbank, 2010).

### 2.2 Sampling & analytical methods

River sediments were sampled from the Tibetan headwaters down to the Brahmaputra in Bangladesh, including the major

tributaries. For sample locations, upstream of the Himalayan front (Pasighat, pt. 8), sediments were taken from fresh





sandbars. Downstream of the Himalayan front, river sediments were dredged in the center of the channel using local boats. A number of sampling points were revisited several years to assess the temporal variability of the [10]Be signal. After collection, samples were dried and sieved into several grain-size classes ranging from 63 to 1000 μm (Table S1).

Grain size fractions were then separated with a Frantz magnetic separator and the non-magnetic fraction was further treated with HCl (38%) to remove carbonates. This fraction subsequently underwent 4 to 5 leachings (3 to 5 days each) in $H_2SiF_6$ (34%) – HCl (38%) (2/3 – 1/3 vol.) alternating agitation and heated ultrasonic bath treatments (70°C). The purified quartz fractions were then further cleaned, to remove possible remaining meteoric [10]Be contributions, through 3 subsequent HF baths in stoichiometric amounts so as to dissolve 10% of the total sample mass at each step. Chemical separation of the

purified quartz was performed in the labs of ISTerre – Grenoble (TSA samples) and at the ETH Zürich (remaining samples). In both cases procedures were very similar. Purified quartz is dissolved in concentrated HF along with a small amount of [9]Be carrier solution (Table S1). Be was separated by ion-exchange chromatography, precipitated as $Be(OH)_2$ and dehydrated to BeO at 1000°C (von Blanckenburg et al., 1996). Systematic procedural blanks were also separated using similar procedures.

[10]Be/[9]Be ratios were measured on different AMS facilities: the 5 MeV ASTER AMS facility at CEREGE (Klein et al., 2008), the 6 MeV TANDEM AMS as well as the 0.5 TANDY AMS of ETH Zürich (Christl et al., 2013). [10]Be/[9]Be ratios measured at the CEREGE were normalized to NIST 27900, with an assumed [10]Be/[9]Be ratio of $2.79 \times 10^{-11}$ while the [10]Be/[9]Be ratios measured at ETH are normalized to ETH standard S2007N, with an assumed ratio $2.81 \times 10^{-11}$ (Christl et al., 2013). Both normalization procedures are comparable and equivalent to the 07KNSTD standardization (Nishiizumi et al.,

2007) with a [10]Be half-life of 1.387 Ma (Chmeleff et al., 2010; Korschinek et al., 2010). Sample AMS measurements were corrected for procedural blanks with [10]Be/[9]Be ratios ranging from $2.35\ (\pm1.07) \times 10^{-15}$ to $7.99\ (\pm2.17) \times 10^{-15}$ depending on measurement batch and date (Table S1).

   Recent work from Portenga et al. (2015) suggests the possible occurrence of [9]Be in the analyzed quartz aliquots despite

careful cleaning and purification of the samples. Ca. 3 g of purified quartz were dissolved in concentrated HF, dried down and re-dissolved in a 2% $HNO_3$ solution. The [9]Be concentration of the solution was measured on a PerkinElmer SCIEX Elan DRC-e quadrupole ICP-MS at the ETH Zürich by standard-sample bracketing. This approach yielded consistent results compared to a subset of samples measured by standard addition, suggesting that ICP matrix effects are limited. The external reproducibility of the [9]Be concentration measurements by standard-sample bracketing was ca. 10% (1 sigma). For samples

with enough material, replicate 3 g quartz splits were measured to assess the homogeneity of the [9]Be concentration among different quartz aliquots.

   We also included in this study the [10]Be data concentration data of large tributaries that were measured by Finnegan et al. (2008) on the upper part of the catchment as well as the Yigong and Parlung Rivers (pts. 3, 4, d and e); by Portenga et al.

(2015) for samples of the Tsang-Chhu (pt. m) and by Abrahami et al. (2016) for samples from the Teesta (pt. n). For consistency, the [10]Be concentration data reported in these publications were re-processed using the same procedure as our data. Existing additional [10]Be data within the Tsangpo-Brahmaputra catchment from Le-Roux-Mallouf et al. (2015) were not included in this study as they mainly focus on smaller, second order catchments within the Bhutan Himalaya.

### 2.3 Catchment wide denudation rate and sediment flux calculations

The procedure for the determination of basin-wide denudation rates using the TCN data is similar to that detailed in Lupker et al. (2012) and is summarized hereafter. All data are reported in Table S1. Basin average denudation rates, $\bar{\varepsilon}$ (cm × yr$^{-1}$),





were calculated from the measured $^{10}$Be concentration, $N$ (at × g$^{-1}$), following Eq. (1) (Brown et al., 1995), thereby assuming efficient mixing of the sediments in the river system and steady state denudation:

$$N = \frac{1}{\bar{\varepsilon}} \times \left( \frac{\overline{P_{n_i}}}{\mu_n} + \frac{\overline{P_{ms_i}}}{\mu_{ms}} + \frac{\overline{P_{mf_i}}}{\mu_{mf}} \right) = \frac{1}{\bar{\varepsilon}} \times \sum_i \frac{\overline{P_i}}{\mu_i}$$

(1)

Where $\overline{P_{n_i}}$, $\overline{P_{ms_i}}$, $\overline{P_{mf_i}}$ are the average basin wide $^{10}$Be production rate (at/g/yr) by neutrons, slow muons and fast muons, respectively, and $\mu_i = \rho/\Lambda_i$ with $\rho$ being the average density of the eroded material (here 2.7 g/cm$^3$) and $\Lambda_i$, the effective attenuation lengths of neutrons, slow, and fast muons ($\Lambda_n \sim 160$, $\Lambda_{\mu s} \sim 1500$ and $\Lambda_{\mu f} \sim 4320$ g/cm$^2$, Braucher et al., 2011). Equation (1) implies that the radioactive decay of $^{10}$Be can be neglected compared to the denudation rate ($\bar{\varepsilon} \times \mu_i \gg \lambda$), which is the case for typical denudation rates in tectonically active areas.

The basin wide $^{10}$Be production rate $\overline{P_i}$ was determined following Eq. (2):

$$\overline{P_i} = \frac{1}{S} \iint_{x,y} P_i(x,y) \times F_{topo}(x,y) \times F_{glacier}(x,y) \times dxdy$$

(2)

where $S$ is the basin area, $P_i(x,y)$ the local production rate at each point of the basin and $F_{topo}$ and $F_{glacier}$ are topographic and glacier correction factors integrated over the basin surface $S$. To calculate $P_i(x,y)$, the sea level high latitude (SLHL) $^{10}$Be production rates by neutrons, slow, and fast muons were scaled for latitude and local altitude according to Lal (1991) / Stone (2000) (St scaling scheme). For this work, we used the most recent nucleonic SLHL production rate of 4.01 ± 0.10 at/g/yr

compiled by Phillips et al. (2016) and the slow and fast muonic contribution of Braucher et al. (2011). Given the high variability of muonic production rates reported in the literature (e.g. Heisinger et al., 2002a; Heisinger, et al., 2002b; Braucher et al., 2003; Braucher et al., 2011; Phillips et al., 2016), we propagated a 50 % uncertainty on the SLHL muonic production rates. Altitude scaling was based on the 3 arc second SRTM-4 DEM dataset (Jarvis et al., 2008). A topographic correction factor ($F_{topo}$) from shielding by surrounding topography was applied on a pixel-by-pixel basis for 10° azimuth

steps and a 15 km surrounding area following Dunne et al. (1999). Production rates under glacial areas were set to zero ($F_{glacier}$) using the Global Land Ice Measurements from Space (GLIMS) glacier database (Raup, 2007). Denudation rates were computed from Eq. (1) and production rates estimated from Eq. (2).

The cosmogenic-derived sediment fluxes ($\varphi_{cosmo}$) were computed from the cosmogenic denudation rates following $\varphi_{cosmo} = \varepsilon$

$\times \rho \times S$, (with $\rho$ the density of the eroded material, 2.7 g/cm$^3$ and S the basin surfaces). For sediments sampled in rivers downstream of the Himalayan front and having a significant upstream Himalayan drainage, we derived an apparent Himalayan denudation rate by restricting the mean production rate to the Himalayan topography, i.e. by considering the area above 300 m a.s.l. (roughly the floodplain to Siwalik transition) (Lupker et al., 2012). It assumes no sediment addition by incision or significant in-situ $^{10}$Be production in the Brahmaputra plain which is supported by the absence of any trace of

significant river incision in the floodplain.

### 3 Results

#### 3.1 Occurrence of "natural" $^9$Be

If unaccounted for, the occurrence of $^9$Be in addition to the amount of $^9$Be introduced by the carrier solution leads to an underestimation of cosmogenic $^{10}$Be concentrations and hence an overestimation of calculated denudation rates. Measured

$^9$Be concentrations are shown in Table S2 and were put into relation to the amount of $^9$Be added to the sample by the carrier



solution. 8 out of the 33 samples showed significant amounts of natural $^9$Be in the quartz samples (i.e. 5% or more of the $^9$Be amount of the carrier) with $^9$Be concentrations in quartz reaching up to 1.5 ppm. It is noteworthy to highlight that high $^9$Be concentrations were mainly found in samples from 3 adjacent catchments, tributaries to the Brahmaputra: the Tsang Chhu (pt. m in Fig. 1), the Manas (pt. l), and the Subansiri (pt. j). While we are not aware of any measurement of $^9$Be in the quartz

from the Manas and the Kameng, the high $^9$Be concentrations reported by Portenga et al. (2015) are all from the Tsang Chhu catchment, draining west Bhutan. This suggests that the occurrence of significant $^9$Be concentrations in quartz from the Brahmaputra basins is restricted to a few basins on the southern edge of the Tibetan Plateau. Replicate analyses show that for samples with a low $^9$Be content, the reproducibility is satisfactory, however the samples with higher concentrations also show a high variability in their $^9$Be content amongst aliquots of the same sample. This suggests that the carrier phase of $^9$Be

is heterogeneously distributed within the sample and is prone to nugget effects. Accurately correcting for this additional $^9$Be contribution, therefore, proves to be difficult based on the *a-posteriori* measurements of $^9$Be in quartz aliquots of the sample. These observations call for the systematic measurement of the $^9$Be concentration in an aliquot of the quartz sample that is being used for the AMS $^{10}$Be/$^9$Be ratio measurement as suggested by Corbett et al. (2016). In the case of this work, we nevertheless made an attempt to correct for the possible occurrence of $^9$Be by propagating conservative uncertainties on the

amounts of natural $^9$Be that were measured by ICP-MS (Table S1). Given the variability in the replicate $^9$Be data (Table S2) we propagated an overall uncertainty of 100% on the measured $^9$Be concentration. In most cases, where the contribution of natural $^9$Be compared to the carrier $^9$Be content is small, this additional uncertainty had little effect on the final uncertainty of the $^{10}$Be concentration. It is only significant for samples with high concentrations of natural $^9$Be from the Tsang Chhu (pt. m), the Manas (pt. l), and the Subansiri (pt. j) (Table S1). Samples for which no natural $^9$Be concentration measurements

could be made did not belong to these basins and therefore likely only contain small amounts of $^9$Be. We nevertheless conservatively propagated an uncertainty of 2% on the total amount of $^9$Be added to these samples.

### 3.2 $^{10}$Be concentrations, denudation rates and erosion fluxes

Overall, the $^{10}$Be concentrations measured across the Tsangpo-Brahmaputra almost span 3 orders of magnitude and vary from $1.2 \times 10^6$ at/g to $5 \times 10^3$ at/g. Concentrations are highest in the upper reaches of the Tibetan plateau and decrease

downstream. The lowest $^{10}$Be concentrations are measured for tributaries of the Tsangpo-Brahmaputra that drain the south and eastern part of the range (Teesta, Subansiri, Dibang and Lohit). The $^{10}$Be concentrations were measured in a total of 4 different grain-size fractions (63-125 μm; 125-250 μm; 250-500 μm; 500-1000 μm) to assess possible grain-size effects across the Tsangpo-Brahmaputra catchment. Samples only contained enough material for $^{10}$Be measurements in two adjacent grain-size classes at most. The difference in $^{10}$Be concentration across grain size classes of a same sample remains limited to

30% (Fig. 2) except for sample BRM1224 (pt. 7) for which the measured 500-1000 μm grain size concentration is a factor of 2 to 3 higher than the finer fraction. No systematic trend with grain-size is observed across the grain sizes that were measured in this study. Concentrations measured on the Tsangpo-Brahmaputra in Bangladesh (pt. 11) are also variable within a factor 2 for the 7 measurements (5 samples) that were performed.

Catchment wide denudation rates were calculated using Eq. (1) and Eq. (2). The denudation rates measured in the Tsangpo-Brahmaputra basin range from 0.03 mm/yr to > 4 mm/yr with generally low denudation rates measured on the Tibetan Plateau (< 0.3 mm/yr). Denudation rates measured along the Tsangpo-Brahmaputra mainstream increase downstream of the NBGPm and reach values of ca. 0.5 to 1.5 mm/yr in the floodplain (floodplain corrected values, i.e. contribution from upstream areas above 300 m a.s.l. only). In Bangladesh, the average denudation rate for the entire Tsangpo-Brahmaputra

catchment is 1.0 ± 0.2 mm/yr based on all measurements. The highest measured denudation rates associated to the rivers draining the southern and especially the eastern flank of the range culminate above 2 mm/yr for the Lohit and Dibang Rivers.





The sediment fluxes that are derived from the denudation rate estimates generally increase downstream along the main stream of the Tsangpo-Brahmaputra and reach 1140 ± 240 Mt/yr at the outlet in Bangladesh. It is noteworthy to point out that the main increase in sediment fluxes occurs during the transit of the river through the Himalayan range and the NBGPm (location 3 to 8) but that the sediment fluxes as measured by [10]Be show only little increase within the floodplain (location 8 to 11) despite some significant inputs from tributaries. The main tributaries to the Tsangpo-Brahmaputra in terms of sediment flux are the Yigong and Parlung Rivers within the syntaxis (combined, ca. 100 Mt/yr) and further downstream the Lohit, Dibang and Subansiri catchments that export each more than 100 Mt/yr based on the [10]Be data.

## 4 Discussion

### 4.1 Evolution of TCN signals along the Tsangpo-Brahmaputra

#### 4.1.1 Denudation rates lag downstream of the NBGPm

The Tsangpo-Brahmaputra catchment has a unique setting in that the main trunk river drains highly contrasted physiographic units with different denudation rates. The NBGPm in particular shows very active erosion processes, with steep river gradients, active landsliding that coincide with rapid exhumation (Finnegan et al., 2008; Stewart et al., 2008; Enkelmann et al., 2011; Larsen and Montgomery, 2012; Lang et al., 2015; King et al., 2016). The NBGPm separates the slowly eroding (< 0.3 mm/yr), high altitude (> 4000 m) Tibetan Plateau from the lower lying reaches of the Tsangpo-Brahmaputra (< 2000 m) and the Indo-Gangetic floodplain (< 200 m). The effect of the NBGPm on TCN concentrations is shown in Fig. 3. The [10]Be concentration of sediments sampled along the mainstream of the Tsangpo-Brahmaputra decreases by almost an order of magnitude downstream of the NBGPm, highlighting the dilution of high [10]Be concentration Tibetan Plateau sediments by low concentration sediments produced in the NBGPm. This decrease in concentration coincides with the river's main knickpoint, which is also the zone of current high erosion rates as documented by the literature.

The Tsangpo-Brahmaputra [10]Be CWDR long-profile in Fig. 3 shows that the denudation rates rapidly increase (from less than 0.1 mm/yr before pt. 3 to ca. 1 mm/yr for pt. 6 to 11) across the eastern Himalayan syntaxis. While discrepancies between consecutive upstream and downstream estimates (pt. 7 and 8) highlight the inherent variability of the sediment transport system, this increase in measured denudation rates supports the progressive addition of rapidly eroding areas to the total drainage area of the main stream Tsangpo-Brahmaputra. But Fig. 3 also suggests that the main increase in denudation occurs in the syntaxis, between pt. 4 and 7, *i.e.* about 150 km downstream of the NBGPm area of high exhumation and denudation rates (Finnegan et al., 2008; Stewart et al., 2008; Enkelmann et al., 2011; Larsen and Montgomery, 2012). Catchment wide denudation rates reach a maximum just before crossing the Main Frontal Thrust (MFT) and exiting the Himalayan range at pt. 8. A similar observation can be drawn from the evolution of the [10]Be derived sediment flux. The measured sediment fluxes remain low even-after crossing the main river knickpoint of the NBGPm with TCN sediment fluxes only increasing significantly after pt. 4. In Fig. 4, [10]Be-derived sediment fluxes are plotted as a function of upstream area. On such a plot, the average denudation rate over a given river reach, is given by the slope between data points. The highest denudation rates (i.e. the steepest slopes between consecutive sampling points on Fig. 4) are documented between pt. 4 and 7, hence downstream of the NBGPm, also pointing towards high apparent denudation rates in the syntaxis, downstream of the NBGPm.

#### 4.1.2 True denudation rates or possible biases in the TCN signal?

The presence of an intensely eroding area in the eastern Himalayan range downstream of the NBGPm, along the lower Siang, down to the MFT, has not been documented by previous studies. Studies suggest that most of the sediment is produced within the NBGPm on both the long and short term: thermochronological data (Finnegan et al., 2008; Stewart et



al., 2008; Bracciali et al., 2016; Salvi et al., in press) display minimum closure ages for various thermochronometers in a reduced area centered on the confluence between the Po-Tsangpo and the Yarlung Tsangpo, even though this high denudation rates area has been suggested to extend more southwards than earlier recognized (Enkelmann et al., 2011); the recent erosional activity displays a similar pattern with a maximum landslide density in the NBGPm around the above

mentioned confluence, and which significantly decrease further downstream, along the lower Siang (Larsen and Montgomery, 2012). Our [10]Be denudation rates do not show any compelling evidence of high denudation along the lower reach of the Siang river, i.e. downstream the NBGPm: the [10]Be denudation rates from tributaries of the main-stream lower Siang such as the Syom River (pt. f) or three other smaller tributaries (pts. p,q and r) indicate apparent denudation rates of 0.4 to 0.6 mm/yr in that area much lower than the 4 to 10 mm/yr suggested by the slope between pts. 4 and 7-8 on Fig.4. In

the absence of other evidence for localized and intense denudation downstream of the NBGPm, we hypothesize that resolving these apparently contradictory results require considering possible biases in the apparent CWDRs along the Tsangpo-Brahmaputra: several scenarios are therefore explored in the following.

Calculations of CWDR most commonly hypothesize that quartz is ubiquitous. However, non-uniform quartz distributions

may yield significant biases in the downstream TCN-derived sediment fluxes (Carretier et al., 2015). In our case, very low quartz concentrations in the outcropping rocks of the NBGPm might explain the apparent absence of response in the CRN-derived flux signal (Fig. 4). If the Tibetan headwaters partly drain carbonate-rich Tethyan Sedimentary Series (TSS) (Liu and Einsele, 1994) or Linzigong volcanics, the quartz flux exported by the Tibetan Plateau would tend to be overestimated. Nevertheless, if we except a thin band of marbles and ophiolites along the highly-deformed suture, the metamorphic or

igneous, gneissic and granulitic lithologies outcropping in the NBGPm do not present, to our knowledge, major difference in quartz amounts compared to the Himalayan metamorphic rocks further south. In addition, such bias would not permit explaining the very high apparent increase in sediment flux along the lower Siang (Fig. 4).

Another possible source of perturbation of TCN signals in high altitude environments such as the Himalayas is the input of

glacial sediments. Glacially sourced sediments have been shielded from cosmic rays due to the thick ice cover and therefore have low [10]Be concentrations (Godard et al., 2012; Delunel et al., 2014). Fig. 5 shows the evolution of the drained glaciated area along the main stream of the Tsangpo-Brahmaputra. The TCN–derived denudation rates increase downstream of the major area of glaciation and the NBGPm. It is therefore unlikely that glacier sediments are the main cause for the observed downstream lag in TCN signals since the glaciated proportion of the catchment is already significant upstream of the main

knickpoint (i.e. upstream of the main increase in CWDR) and increases little downstream of it. The main increase in glacial contribution occurs in the NBGPm region, as the Tsangpo receives the contribution of the heavily glaciated Yarlung-Parlung catchments and hence would result in a decrease in [10]Be concentrations within the NBGPm rather than further downstream.

Additional processes that may affect TCN signals in steep, highly eroding areas are stochastic events and poor sediment

mixing. Landslides and other catastrophic events may perturb the downstream TCN concentration and hence bias calculated CWDRs by supplying low TCN concentration sediments from previously shielded bedrock (Kober et al., 2012). Modelling studies suggest that this effect is more pronounced for small catchments of a few tens of $km^2$ or less (Niemi et al., 2005; Yanites et al., 2009). As the catchments drained by the Tsangpo-Brahmaputra downstream of the NBGPm exceed 200000 $km^2$, it is unlikely that regular and diffuse landsliding would significantly affect the CWDRs. The differences in [10]Be

concentration and CWDR obtained from sediments at pt. 7, 8 and 9 nevertheless suggest that stochastic processes might influence the sedimentary signal. There is no tributary upstream of pt 8. that could explain the anomalously high [10]Be concentration (or anomalously low denudation rate) obtained for samples from pt 8. The reasons for this particular perturbation remain unclear but the river reach between the NBGPm and the MFT is prone to extreme hydrological and





erosion events (Lang et al., 2011). A recent major perturbation of the sedimentary system was provided by the catastrophic breaching of a landslide-dammed lake on the Yigong River in 2000, which resulted in a devastating flood downstream of the confluence with the Tsangpo-Brahmaputra (Delaney & Evans, 2015). The high discharge that followed the dam breach, mobilized about $1 \times 10^{8}$ m³ or 270 Mt of landslide material in the area (Larsen and Montgomery, 2012). This additional flux

of sediments is equivalent to the amount of sediments transported by the Tsangpo in 1 to 3 years depending on the sediment flux estimates. Some remnant sediments of this event are still present in the channel downstream of the breach (Lang et al., 2013), but over 12 years after this catastrophic flood it is unlikely to they still dominate the sediment budget at the time of sampling.

In the absence of clear geomorphological indications that denudation rates downstream of the NBGPm are actually higher than the in the NBGPm and in the absence of evidence for other perturbations of the TCN signal such as glacial sediment input, diffuse landsliding or recent catastrophic flood events that could explain the lag in denudation rates downstream of the NBGPm, other processes have to be considered.

**4.1.3 Abrasion of landslide material**

Abrasion of the sediment load during fluvial transport can affect TCN concentrations since the grain-size fraction analyzed is not necessarily representative of the entire sedimentary load and different grain-sizes may have been affected by varying erosion and transport processes (Carretier & Regard, 2011). Olen et al. (2015) suggest that abrasion of the sand fraction during transport could affect the TCN signal in some Himalayan rivers, when sand grains produced in the headwater regions

are abraded during transport and transferred to grain-sizes smaller than what is typically analyzed. However, in the case of the Tsangpo-Brahmaputra this would imply that sediments coming from the South Tibetan region would pass below the analyzed grain-size fraction and be overwhelmed by low [10]Be concentration sandy material from the NBGPm, which is the opposite of what we observe. In addition, once within the sand fraction, abrasion processes are limited by viscous dampening during transport (Jerolmack and Brzinski, 2010) and the downstream fining observed in sand dominated rivers occurs over

very long distances (Frings et al., 2008). Abrasion processes during fluvial transport may also play a role in the TCN concentration evolution of sandy material through the production of sand by pebble attrition. In the case of the NBGPm, landsliding is the main process that produces and transfers sediments to the main channel (Larsen and Montgomery, 2012). Landslides predominantly produce coarse debris (Attal and Lavé, 2006) and can perturb downstream signals (Kober et al., 2012; West et al., 2014). The large amount of coarse debris delivered to the river is however only reflected in the TCN

concentrations of the sand fraction once they are transferred into grain-sizes (between 125 to 1000 µm) due to abrasion (Fig. 6). In the case of the NBGPm, we therefore suggest that the upstream, high TCN concentration signal from the Tibetan Plateau could potentially be only slowly diluted by low concentration sediments produced in the NBGPm because of the transport distance necessary to transform landslide clasts into sandy material by attrition.

To explore the effects of abrasion on the TCN signal in the Tsangpo-Brahmaputra catchment, we constructed a quantitative model of fluvial abrasion that includes the dominant landslide-derived sediment production (Attal and Lavé, 2006). This model allows us to estimate the typical distances over which this dilution is predicted to occur in a setting such as the Tsangpo-Brahmaputra. The model evaluates the response of apparent TCN denudation rates, $\bar{\varepsilon}_{app}$ , for a given input sediment flux and TCN concentration that is progressively diluted by the abrasion products of coarse landslide material.

Each block or rock fragment, after delivery from the hillslopes to the river network, will be submitted to breaking, crushing and abrasion that tend to round the fragment and decrease its diameter. We assume that pebbles >2mm in diameter are abraded over a distance $dL$ following the commonly used Sternberg's law (Sternberg, 1875), $dV/V = -k\, dL$, with $k$ the



pebble abrasion coefficient (in km$^{-1}$), $V$ the pebble volume and L the position along the channel. The products of abrasion are mostly fine sediments that then transit as suspended load (Kuenen, 1956) and may be collected for a typical TCN sample (125 – 1000 μm). In addition to the abrasion coefficient, $k$, the main parameters controlling the delivery and evolution of the sediment load are: $f_g$, the initial fraction of landslide-derived material that is delivered to the channel as pebbles and larger

clasts; $f_{as}$, the fraction of abrasion products produced within the channel that is transferred from the coarser fraction to a grain size sampled for TCN analyses (typically 125 – 1000 μm); and $f_{TCN}$ the initial fraction of landslide material directly delivered to channel in the TCN grain-size fraction.

The volumetric flux of sediments within the TCN grain-size fraction ($Q_{TCN}$) that is transported at a given distance $L$ from the
NBGPm (pt. 3) along the channel is given by the sum of: i) TCN-sized sand from Tibetan Plateau, $Q_{T,TCN}$, ii) abrasion of bedload delivered from the Tibetan Plateau, $Q_{T,b}$, iii) direct delivery of TCN-sized sand from landslides within the highly eroding areas, and, iv) the abrasion of coarse landslide material, therefore:

$$Q_{TCN}(L) = Q_{T,TCN} + Q_{T,b}f_{as}(1-e^{-kL}) + f_{TCN}\int_0^L w(x)\varepsilon(x)\,dx + f_g f_{as}\int_0^L w(x)\varepsilon(x)\left[1-e^{-k(L-x)}\right]dx \tag{3}$$

where $w(x)$ is the local catchment width and $\varepsilon(x)$ the local denudation rate.

The average TCN concentration of sediments along the main channel, $N(L)$, is then expressed as:

$$N(L) = \frac{N_T\left[Q_{T,TCN}+Q_{T,b}f_{as}(1-e^{-kL})\right]+f_{TCN}\int_0^L N(x)w(x)\varepsilon(x)dx+f_g f_{as}\int_0^L N(x)w(x)\varepsilon(x)\left[1-e^{-k(L-x)}\right]dx}{Q_{T,TCN}+Q_{T,b}f_{as}(1-e^{-kL})+f_{TCN}\int_0^L w(x)\varepsilon(x)dx+f_g f_{as}\int_0^L w(x)\varepsilon(x)\left[1-e^{-k(L-x)}\right]dx} \tag{4}$$

In steady state, the sediment TCN concentration is related to denudation through Eq. (1), so that Eq. (4) can be re-written as:

$$N(L) = \frac{N_T\left[Q_{T,TCN}+Q_{T,b}f_{as}(1-e^{-kL})\right]+f_{TCN}\int_0^L \sum_i \frac{P_i(x)}{\mu_i}w(x)dx+f_g f_{as}\int_0^L \sum_i \frac{P_i(x)}{\mu_i}w(x)\left[1-e^{-k(L-x)}\right]dx}{Q_{T,TCN}+Q_{T,b}f_{as}(1-e^{-kL})+f_{TCN}\int_0^L w(x)\varepsilon(x)dx+f_g f_{as}\int_0^L w(x)\varepsilon(x)\left[1-e^{-k(L-x)}\right]dx} \tag{5}$$

It is reasonable to assume that the upstream coarse bedload sediment flux is limited because of slow denudation rates, long travel distances, that allow efficient abrasion and possible sediment trapping on the Tibetan Plateau. This is supported by the valley infill upstream of the gorges being dominated by sands over the upper 100 to 200 m (Wang et al., 2014). Equation (5)
simplifies to:

$$N(L) = \frac{N_T Q_{T,TCN}+f_{TCN}\int_0^L \sum_i \frac{P_i(x)}{\mu_i}w(x)dx+f_g f_{as}\int_0^L \sum_i \frac{P_i(x)}{\mu_i}w(x)\left[1-e^{-k(L-x)}\right]dx}{Q_{T,TCN}+f_{TCN}\int_0^L w(x)\varepsilon(x)dx+f_g f_{as}\int_0^L w(x)\varepsilon(x)\left[1-e^{-k(L-x)}\right]dx} \tag{6}$$

The TCN concentration can then be converted to an apparent denudation rate using Eq. (1) applied to the whole drainage
area, including the southern Tibetan part:

$$\bar{\varepsilon}_{app} = \frac{1}{N(L)}\cdot\sum_i \frac{\bar{P}_i(L)}{\mu_i} = \frac{1}{N(L)}\cdot\frac{1}{S}\int_{\substack{upper\\drainage}} \sum_i \frac{\bar{P}_i(x)}{\mu_i}w(x)dx \tag{7}$$





In order to test this abrasion model, the river section downstream of the Tibetan Plateau is simplified to a 500 km long reach with a constant width of 100 km, yielding a total catchment area similar to that of the actual catchment between pt. 3 and 8. (Fig. 7a). The $^{10}$Be production rates in our model are also set to similar values as the average production rates modeled for the Tsangpo-Brahmaputra between pt. 3 and 8 (for neutron and muon production scaling factors of 12.7 and 4.4

respectively). Sediment is delivered from upstream with a $^{10}$Be concentration of ca. $6.5 \times 10^5$ at/g and a total flux of 40 Mt/yr, similar to the sediment flux delivered by the Tibetan Plateau (based on the average concentration at pt. 3). The fraction of this flux within the TCN grain-size assumes that all coarse material has been abraded: $Q_{T,TCN} = (f_{TCN} + f_g f_{as})Q_T$. The 500 km reach is further sub-divided into a 200 km long upstream area with high denudation rates, $\varepsilon_{upstr}$, representing the supposedly rapid exhumation and erosion NBGPm area followed by a 300 km long lower catchment with

lower denudation rates $\varepsilon_{dwnstr}$. Different pairs of denudation rates can be chosen but if $\varepsilon_{dwnstr}$ is set to typical background Himalayan denudation rates of ca. 1 mm/yr (Lupker et al., 2012), $\varepsilon_{upstr}$ needs to range between ca. 5 and 12 mm/yr to satisfy the actual measured downstream $^{10}$Be concentration (ca. $3.9 - 7.8 \times 10^4$ at/g). The range of abrasion parameters, $k, f_g$, $f_a$ and $f_{TCN}$ are estimated from abrasion experiments and grain-size determinations of landslide material (Attal & Lavé, 2006; 2009; Nibourel et al., 2015).

Fig. 7 shows the apparent $^{10}$Be denudation rate in the case of abrasion and for a range of model parameters in comparison to the no abrasion case. These model results show that for the range of parameters considered, there is a downstream spatial lag between the actual denudation rates (no abrasion case) and the denudation rates determined by the $^{10}$Be concentrations of sediments found in the main channel. The magnitude of this lag is dependent on parameters but ranges from ca. 50 to 150

km. This abrasion model is sensitive to the magnitude of denudation rates in the rapidly eroding part ($\varepsilon_{upstr}$, Fig. 7b) and, in the case of low denudation rates in the Tsangpo it is unlikely to be detected, as the difference between the denudation rates reconstructed for the abrasion and non-abrasion cases are very similar. The magnitude of the lag is also dependent on the chosen abrasion coefficient. As $k$ increases, the transfer of mass from the low concentration landslide material to the analysed grain size is more rapid and hence the dilution of high concentration upstream sediments occurs earlier along the

river reach. However, Fig. 7 also shows that the downstream lag is not sensitive to changes in $f_{TCN}$, $f_g$ and $f_{as}$.

These model results suggest that abrasion of landslide material is able to induce a downstream lag in the denudation rate calculated from $^{10}$Be concentration in the sand fraction. The magnitude of the modelled offset is within the range of the observed offset between the supposedly high denudation rate area within the NBGPm and the maximum denudation

measured occurring about 150 km further downstream. However, it should also be noted that our model likely over-estimates the importance of abrasion since part of the sediment load has already been transported over some distance before it is delivered to the main channel and hence has already experienced abrasion.

On the other hand, the absence of relation between grain-size and $^{10}$Be denudation rate within the sand size fraction (Fig. 2)

might seem at odd with our argument that abrasion processes play an important role in the transfer of the erosional signal from hillslopes to river sediments. Identifying a potential contradiction, and more largely predicting a relation between grain-size and TCN concentration all along the Siang, would however require documenting in details the grain size distribution in the landslides and in the abrasion products, and how these depend on bedrock weathering intensity, climatic, lithologic and tectonic setting, or on hydrodynamic conditions during transport, respectively. For the moment, the absence of

database or theoretical models on these processes prevents detailed exploration of the sand size issue. Our simplified model could actually only be validated or invalidated by comparing $^{10}$Be concentration across a large range of grain-sizes, i.e. between sand size content and boulder size content.





Abrasion effects in the Tsangpo-Brahmaputra are unlikely to be restricted to detrital TCN studies. Other detrital proxies such as thermochonological ages of sediment grains should be affected similarly, i.e., the young cooling ages found in the NBGPm (e.g. Zeitler et al., 2014) should be transferred to the typically studied sand fraction of the sediment load further downstream only after coarse landslide material is abraded. The population of Zircon fission track (FT) ages (Enkelmann et

al., 2011) and $^{39}Ar/^{40}Ar$ ages (Lang et al., 2016) were measured in three Tsangpo-Brahmaputra river sediment samples from downstream of the NBGPm to the Himalayan front (Fig. 8). In both cases, the relative proportion of the youngest grains (< 2 Myr for Zircon Ft and < 4 Myr for $^{39}Ar/^{40}Ar$) is constant or slightly increases from up to downstream. Even though, the increase is small and needs to be interpreted carefully given the limited number of samples and the high variability of the system, we would rather expect a decrease in the abundance of the young NBGPm-derived grains as sediments eroded in the

NBGPm are progressively diluted by older grains eroded from the slower exhumation rate areas in the syntaxis, downstream of the NBGPm. This small increase in the proportion of young grains rather supports our suggestion that sediments produced in the NBGPm are only transferred to the sand fraction of the Tsangpo-Brahmaputra further downstream due to abrasion processes. This observation suggests that care should be taken before relating detrital signals to spatial patterns of denudation within the NBGPm and that abrasion should be considered in actively eroding areas for a range of detrital

sediment provenance and denudation tracers.

### 4.2 Denudation budget of the Tsangpo-Brahmaputra

### 4.2.1 Denudation of the upstream Tibetan plateau part of the Tsangpo-Brahmaputra catchment

Denudation rates measured upstream of the NBGPm and on the Tibetan plateau are the lowest denudation rates measured in the Tsangpo-Brahmaputra catchment (ca. 0.04 to 0.2 mm/yr). These low denudation rates are of the same order of magnitude

as previously published bedrock and detrital denudation rates in other areas of the plateau (Lal et al., 2004; Kong et al., 2007; Hetzel et al., 2011; Strobl et al., 2012; Li et al., 2014; Rades et al., 2015) The slightly higher denudation rates measured for the two northern tributaries, the Kyi (pt. a) and Nyang (pt. b) (0.08 and 0.2 mm/yr, respectively) are associated with higher catchment-averaged relief and slopes (Table S1). Duplicate measurements from samples taken 4 years apart (this study and Finnegan et al., 2008) show reproducible denudation rates for the Nyang (pt. b), but vary by a factor of two for the

mainstream Brahmaputra-Tsangpo at pt. 3. This later variability could be associated to the stochasticity of the sediment transport system that does not always result in a perfectly mixed sample. It should however be noted that sample TSA-16 yields sediment fluxes of ca. 30 Mt/yr that is compatible with the upstream mass balance (Tsangpo-Brahmaputra at pt.2 with 20 Mt/yr and the Nyang with 10 Mt/yr) while NB-8-04 potentially over-estimates this mass balance (with a flux at pt. 3 of 70 Mt/yr). Overall, the $^{10}Be$ data added by this study highlight the relative landscape stability of the upper part of the Tsangpo-

Brahmaputra catchment in comparison to the Tibetan plateau margins. It also suggests that the sediment supply from the upper reaches of the Tsangpo-Brahmaputra is limited compared to its downstream drained area.

### 4.2.2 Denudation of the eastern syntaxis and of the NBGPm

Denudation rates downstream of the Tibetan plateau, from the NBGPm to the MFT (i.e. the Himalayan part of the Tsangpo-Brahmaputra main stream course) needs to be considered carefully because of the abrasion processes discussed above.

Endorsing this model of downstream lag in the TCN response due to abrasion effects means that the samples along the main stream Tsangpo-Brahmaputra cannot be used to provide tight spatial constraints on the location and magnitude of denudation within the NBGPm. The denudation signal is not immediately transferred to the TCN grain-size in the trunk stream and hence TCN samples do not accurately represent the upstream denudation processes. A first order estimate of the denudation rates in the eastern Himalayan part of the Tsangpo-Brahmaputra can nevertheless be made by assuming that the onset of the

steep gorges and the knickpoint marks the onset of intense landsliding (Larsen and Montgomery, 2012) and that abrasion processes become less significant beyond the gravel-sand transition at the Himalayan front (Dubille and Lavé, 2015), that we





observed to be a few tens of kilometers downstream of the MFT according to alluvial bars exposed along the Tsangpo-Brahmaputra (pt. 8 on Fig. 1). The evolution of the $^{10}$Be-derived sediment fluxes as a function of upstream area (Fig. 4) constrains the average denudation to be within ca. 2.6 to 5.7 mm/yr for that part of the catchment by using the average $^{10}$Be-derived sediment flux of pt. 3 at the gorge entrance and pt. 7 or 8 at the downstream end (calculated using the average

concentration at each point). This estimate is similar to the one made using a $^{10}$Be mixing model (Fig. 9a): the $^{10}$Be concentration of sediments exported at the outlet of the Himalayan front (pt. 8) is calculated by mixing sediments fed by the Tibetan part of the catchment (above pt. 3: 650000 at/g) with sediments produced within the entire syntaxis for a range of denudation rates (for neutron and muon production scaling factors of 12.7 and 4.4 respectively). The downstream $^{10}$Be concentration of sediments that exit the syntaxis should fall within the actual range of measured $^{10}$Be concentrations to be

compatible with our data (3.9 − 7.8 × 10$^4$ at/g) and allows us to constrain the average denudation rates within the entire syntaxis. However, theromochronological data (e.g. Enkelmann et al., 2011; Salvi et al., in press), river gradients (Finnegan et al., 2008) and landslide density (Larsen and Montgomery, 2012) argue for non-uniform denudation in the eastern syntaxis with focused erosion in the NBGPm. A second estimate can therefore be made by assuming that the high denudation rate in the NBGPm is restrained to ca. 5000 km$^2$ (Larsen and Montgomery, 2012). Using a similar mixing approach as exposed

above, we calculated the $^{10}$Be concentration of sediments at the Himalayan outlet that results from the mixing of a) sediments shed by the Tibetan plateau part of the catchment, b) sediments from the NBGPm (with neutron and muon scaling factors of 9.1 and 3.6 respectively) experiencing variable denudation rates ranging from 0 to 50 mm/yr, sediments, c) sediment contributions based on the available $^{10}$Be data from the Rong Chu, Yigong and Parlung Rivers (pts. c, d and e, 130 Mt/yr with a flux weighted average $^{10}$Be concentration of 2.1 × 10$^4$ at/g) and d) sediment contributions from the lower part of

the syntaxis (downstream of the NBGPm) where we assume typical Himalayan denudation rates of 1 to 2 mm/yr. This model shows that the denudation rate within the NBGPm needs to range between ca. 12 and 31 mm/yr to match the actual $^{10}$Be concentration measured at the outlet of the Himalayan front (solid and dashed curve on Fig 9b, depending on the denudation rate assumed for the lower syntaxis). This later estimate is obviously dependent on the denudation rate attributed to the downstream part of the reach, which has not been systematically constrained in this study. Our samples from small

tributaries of the Tsangpo-Brahmaputra suggest denudation rates as low as 0.4 to 0.6 mm/yr (pts. p, q, r and f on Fig. 1) but thermochronological data (Salvi et al., in press) as well as the topographic signature suggest that this area is most probably not experiencing significantly higher denudation rates.

However, one important underlying assumption for these denudation rate estimates, is that the entire upper Tibetan part of

the catchment is connected to the Tsangpo-Brahmaputra trunk stream and that its entire drainage area exports sediments at the calculated rate. A number of observations tend to suggest that this may not the case. The upper Tsangpo is partially dammed along its course by several active and potentially subsiding N-S horst and graben systems (Armijo et al., 1986) and several hundred meters of Quaternary sediments are also stored directly upstream of the Tsangpo Gorge (Wang et al., 2015). Internally drained areas and lakes represent additional breaks in the sediment cascade. Furthermore, the Tibetan headwaters

drain the carbonate-rich Tethyan Sedimentary Series (TSS) (Liu and Einsele, 1994), which contain lower amounts of quartz than the Himalayan or NBGPm metamorphic rocks. This would tend to overestimate the quartz flux that is exported by the Tibetan Plateau relative to the quartz fluxes eroded downstream. In both cases, the actual sediment flux exported by the South Tibetan part of the catchment may be overestimated if calculated directly using $^{10}$Be denudation rates and total catchment area. it would tend to overestimate the quartz flux that is exported by the Tibetan Plateau relative to the quartz

fluxes eroded downstream. Constraining the magnitude of this overestimation requires independent estimate on the sediment flux at the range outlet. Goswami et al. (1985) reported a sediment flux of ca. 210 Mt/yr in Pasighat (pt. 8), which is only 30 to 55 % of the sediment flux estimated from the $^{10}$Be data at the range outlet, and which would correspond to a Tibetan Plateau contribution reduced to only 15 to 40% of the reference flux. However, the sediment gaging details and method of




this flux estimate remain undocumented, and, the time-scale covered by this gaging data is significantly lower than that of the TCN estimates. The TCN mass balance, just upstream of the eastern syntaxis between the Yarlung Tsangpo and the Nyang river, which is unaffected by sediment sequestration in grabens or contrasted quartz contents, is also not very conclusive: the TCN concentration of sample TSA-16 below the confluence of these two rivers (pt. 3) requires mixing

Nyang sediment (pt. b) with 100% of $^{10}$Be-derived (pt. 2) Yarlung Tsangpo sediments flux whereas the sample NB-8-04 rather suggests an upper Tsangpo contribution limited to 20 % of the $^{10}$Be-derived sediment flux. On the other hand, direct estimate of the average sedimentation in Yarlung Tsangpo buried canyon (Wang et al., 2015) over the last 2 Myr suggests that sequestration flux there was limited to ~1 %. Similar calculation for the grabens crossed by the Yarlung Tsangpo further West are not possible due to limited data on subsidence rates: a priori Late Cenozoic low slip rates (<1 mm/yr) on the nearby

Kung Co half graben (Mahéo et al., 2007) would suggest a moderate sequestration of a few percent of the total sediment flux. Furthermore, the modest decrease in TCN concentration between pt. 1 and pt. 2 (Fig. 1) also tends to suggest that sediment sequestration between these points is limited, as otherwise, we would expect the TCN concentration in the mainstream to be more significantly lowered by the lower $^{10}$Be concentration sediment input from the Kyi (pt. a). Finally, the mineralogical composition of the upper Tsangpo-Brahmaputra sediments suggests a lower proportion of quartz grains

compared to those of the Tsangpo-Brahmaputra river close to the outlet but this deficit ranges between 15 and 35 % (Garzanti et al., 2004). Altogether, we therefore suspect the sediment entering into the eastern syntaxis at pt. (3) to be lower than the actual flux predicted by the TCN but we think that this different remains relatively small. We conservatively speculate here, that the actual flux exported by the Tibetan plateau part of the catchment ranges between 50 and 90 % of the flux that would be calculated by the TCN. Using a similar model as exposed above we estimated the effect of a sediment

flux exported by the Tibetan plateau of the catchment that is lower by 50 to 90% compared to the predicted flux (i.e. a sediment flux of 20 to 36 Mt/yr, red envelope on Fig 9a and b). Lowering the upstream sediment flux greatly reduces the denudation rate estimate within eastern syntaxis and the NBGPm (Fig. 9). The denudation estimates, using a reduced Tibetan contribution, ranges from ca. 2 to 5 mm/yr for the average denudation rate estimated over the entire syntaxis (pt. 3 to 3-8) and 4 to 28 mm/yr for the NBGPm (Fig. 9). This later estimate lays at the higher end of the previous estimates of denudation

in the NBGPm, that range between 5 to 17 mm/yr (Stewart et al., 2008; Enkelmann et al., 2011; Larsen and Montgomery, 2012). These denudation rate estimates are also all significantly higher than other $^{10}$Be denudation rates reported for other Himalayan catchments (Vance et al., 2003; Wobus et al., 2005; Finnegan et al., 2008; Lupker et al., 2012; Godard et al., 2012; Godard et al., 2014; Puchol et al., 2014; Scherler et al., 2014; Le-Roux-Mallouf et al., 2015; Portenga et al., 2015; Morell et al., 2015; Olen et al., 2015; Abrahami et al., 2016; Olen et al., 2016) highlighting the significance of the NBGPm

as denudational hot-spot.

### 4.2.3 Downstream denudation of the Tsangpo-Brahmaputra catchment and other Himalayan tributaries

Downstream of the MFT, the Tsangpo-Brahmaputra enters the low-gradient Indo-Gangetic flood plain and receives sediment from a number of significant Himalayan tributaries with denudation rates ranging from 0.7 up to 4 mm/yr. Three main tributaries dominate the sediment input: the Dibang, Lohit and Subansiri (respectively pt. g, h and j in Fig. 1). To our

knowledge, no direct gauging data is available for these rivers, but a significant input from the eastern Rivers (chiefly Lohit and Dibang) was already suggested by the geochemical and mineralogical fingerprinting of sediments in the Tsangpo-Brahmaputra main stream (Singh & France-Lanord, 2002; Garzanti et al., 2004). The Lohit and Dibang catchments are characterized by steep slopes and high relief (Table S1). Furthermore, these catchments receive intense rainfall, in excess of 2 m during the Indian summer monsoon according to remote sensing data (Anders et al., 2006; Bookhagen et al., 2006). GPS

data further suggests rapid convergence across the south-east north-west trending Mishmi and Lohit thrusts (Gupta et al., 2015). The interactions between climate and tectonics in the Lohit and Dibang catchments may therefore be translated into high denudation rates and sediment fluxes. The denudation rates in these catchments have likely been underestimated in the




past. The high denudation rates measured in the Subansiri are more difficult to explain, especially since its neighboring catchments of the Syom (pt. f), the Kameng (pt. k) and the Manas (pt. l) show significantly lower denudation rates for similar catchment average relief and slope (Table S1), closer to denudation rates found elsewhere for large catchments of the Himalaya (Lupker et al., 2012). The Buri Dihing River is the only southern tributary to the Tsangpo-Brahmaputra for which

denudation rates have been measured in this work. The Bhuri Dihing drains the northern part of the Indo-Burmese ranges and denudes at a lower rate (0.2 mm/yr) compared to the main Himalayan tributaries. Denudation rates of the actively uplifting Shillong Plateau have recently been suggested to be very low on average (Rosenkranz et al., 2016).

Overall, according to our TCN data, the Tsangpo-Brahmaputra receives ca. 600 Mt/yr of sediments from its tributaries in the

Indo-Gangetic floodplain downstream of the MFT. Fig. 10 shows the evolution of the sediment flux predicted for the Tsangpo-Brahmaputra by the successive addition of sediment from all its tributaries in the floodplain. Such an approach is only possible for the downstream part of the catchment as our [10]Be data covers all major tributaries of the Tsangpo-Brahmaputra and no significant sediment contribution is considered to come from the floodplain area itself. The predicted flux at the outlet in Bangladesh ranges from 1000 to 1400 Mt/yr depending on the sediment flux that is considered to enter

the floodplain at the MFT (flux estimate of pt. 7 or 8). Fig. 10 shows that the sediment fluxes and denudation rates measured at the outlet in Bangladesh (pt. 11) and in Tezpur in India (pt. 10) are coherent within uncertainty with the simple addition of sediments from the Tsangpo-Brahmaputra. However, the flux predicted at pt. 9 clearly exceeds what can be expected by the sole addition of sediments from the Lohit and Dibang to the sediment load shed by the Tsangpo-Brahmaputra at the MFT (pts. 7 or 8). The [10]Be concentrations measured at pt. 9 may therefore be affected by transient perturbations in the sediment

load or poor mixing of upstream contributions and may not be representative of the average sediment load transported by the Tsangpo-Brahmaputra. Repeated sampling at this location would provide better constraints on the variability of the signal at that location.

[10]Be denudation rates measured in Bangladesh at the outlet of the Tsangpo-Brahmaputra catchment (pt. 11) vary between 0.7

and 1.2 mm/yr for samples that have been collected over 4 different years. This variability is higher than the variability observed for the neighboring Ganga River (Lupker et al., 2012), which suggests that the Tsangpo-Brahmaputra floodplain may be less efficient at smoothing the variability of the upstream TCN signal. The absolute [10]Be sediment fluxes (780 – 1430 Mt/yr, calculated for a catchment above 300 m elevation) are higher by a factor of 1.5 to 2.5 compared to the gaging sediment flux estimates of 500 and 610 Mt/yr (Delft Hydraulics, 1996). A similar overestimation, although of lower

magnitude, was observed for the Ganga catchment (Lupker et al., 2012). A number of factors may affect this comparison: first, the yearly to decadal integration time of gauged fluxes may not be directly compared to the centennial to millennial fluxes estimated from TCN (Wittmann and von Blanckenburg, 2016); second, the subsiding Indo-Gangetic floodplain may be a significant sediment sink, lowering the actual sediment flux exported to the Bay of Bengal; and third, the overestimation of the contributing Tibetan area will also lead to an overestimation the [10]Be sediment flux estimates at the outlet by affecting

sediment contributing area and catchment averaged production rates. The respective contribution of these effects remains difficult to quantify and would warrant a more in depth study, but they need to be considered before directly comparing [10]Be and gauged sediment fluxes in large scale catchments.

## 5   Conclusions

The [10]Be concentration carried by the Tsangpo-Brahmaputra are marked by a sharp decrease over the course of the river.

High [10]Be concentrations, indicative of low denudation rates on the Tibetan Plateau, decrease by almost an order of magnitude as the river crosses the Namche Barwa-Gyala Peri massif (NBGPm). This observation is compatible with the



addition of large amounts of low $^{10}$Be concentration sediments in this region with documented high exhumation rates and active landsliding. We however find that the dilution of the $^{10}$Be signal occurs only about 150 km downstream of the NBGPm. A possible explanation for this downstream lag in response to high denudation is the role of abrasion processes of predominantly coarse material the is delivered by landslides within the NBGPm. High concentration sediment exported by

the Tibetan Plateau is only diluted by low concentration landslide material when abrasion reduces the grain-size of landslide material to a size that falls within the class that is typically analyzed for cosmogenic nuclides. We modeled the effect of abrasion of landslide material on cosmogenic nuclide denudation rates using an idealized reach with similar characteristics as the Tsangpo-Brahmaputra and literature derived abrasion parameters. This model is able to reproduce a downstream lag in the increase of $^{10}$Be-derived denudation rates ranging between 50 to 150 km and is therefore compatible with our data. We

therefore propose that dominant hillslope erosion by landsliding combined with delayed sand production by pebble abrasion might significantly bias TCN-derived denudation rates as commonly measured in the sandy fraction of alluvial sediment load. The effect of fluvial abrasion of landslide material may operate in a large number of actively eroding settings, where landslide is a major sediment production process. Its effect on TCN may be substantial and needs to be taken into account, thereby adding to the limitations of the use of TCN in landslide dominated landscapes, independently of catchment size.

These abrasion effects are also very likely transposable to other detrital sediment studies operationally focused on sand-sized sediments and need to be taken into account for provenance, thermochronological and TCN studies alike.

Our data also provides new constraints on the denudation across the Tsangpo-Brahmaputra catchment. The upstream Tibetan plateau part of the catchment is denuding at a slow rate inferior to 0.2 mm/yr. The observed downstream lag in the

denudation signal of the NBGPm means that the denudation rates within that area cannot be reconstructed at a high spatial resolution. However, we show that it is still possible to use the $^{10}$Be concentrations of samples collected along the main stream of the Tsangpo-Brahmaputra to derive denudation estimates in the eastern Himalayan syntaxis and the NBGPm. The estimate of these denudation rates is however affected by the actual sediment flux exported by the Tibetan plateau of the catchment. Sediment sequestration, disconnected parts of the catchment and lower quartz content of this Tibetan part will

lead to the overestimation of downstream denudation rate estimates. Taking into account these effects we estimate that the denudation rates over the entire syntaxis ranges between ca. 2 to 5 mm/yr. If this denudation is mostly attributed to the rapid denudation rate in the restricted area with rapid exhumation and active landsliding, our $^{10}$Be data suggests denudation rates of 4 to 28 mm/yr. Our data therefore confirms the intense denudation rates in the eastern Himalayan syntaxis region in comparison with other estimates across the Himalayan range. It also highlights the importance of takin into account the

processes in the upstream parts of such a large catchment to derive robust estimates. Finally, the denudation rates of the Tsangpo-Brahmaputra, in the downstream floodplain part of the catchment, are mostly compatible with the addition of fluxes from the Himalayan tributaries and sediments from the outlet in Bangladesh constrain the average denudation rate of the entire Tsangpo-Brahmaputra to be within 0.7 to 1.2 mm/yr.

## 6    Competing interests

The authors declare that they have no conflict of interest.

## 7    Acknowledgments

K. Hippe, N. Hagiphour and S. Gallen are thanked for the fruitful discussions and help in the lab. I. Schimmelpfennig is thanked for her help with the collection of the TSA samples. N. Vögeli is thanked for providing sample BRM MANAS. R.




Braucher and the ASTER team are acknowledged for the swift measurement of TSA-16. M.L. was support by the ETH postdoctoral fellowship program. J.L. and C.F.L. were supported by the ANR Calimero project.

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






**Figure 1: (a) – Overview of the study area. (b) – Shaded-relief map of the Tsangpo-Brahmaputra map with sampling locations of the trunk stream (open symbols) and main tributaries (filled symbols). The NBGPm area characterized by high erosion and fluvial incision rates (> 3 mm/yr) is highlighted in red (see text for references). [10]Be concentrations (c), [10]Be-derived denudation rates (d), [10]Be-derived sediment fluxes (e) of Tsangpo-Brahmaputra mainstream and tributaries. Downstream samples have been corrected for floodplain area by excluding areas below 300 m.a.s.l. from the calculation. Literature data from Finnegan et al. (2008) ([a]), Portenga et al. (2015) ([b]) and Abrahami et al. (2016) ([c]) are also included.**





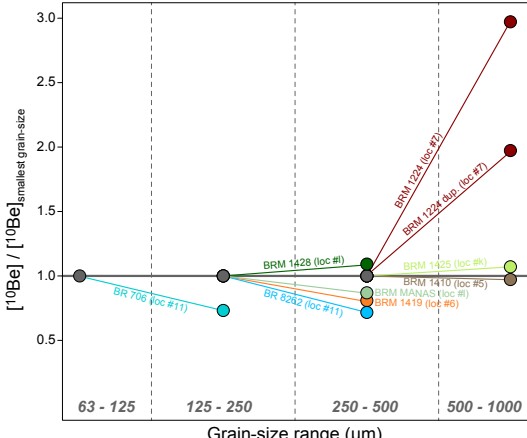

**Figure 2:** Ratio of [10]Be concentrations from grain-size pairs of a same sediment sample. Concentrations are normalized to the concentration of the lowest grain-size fraction for comparison amongst samples.

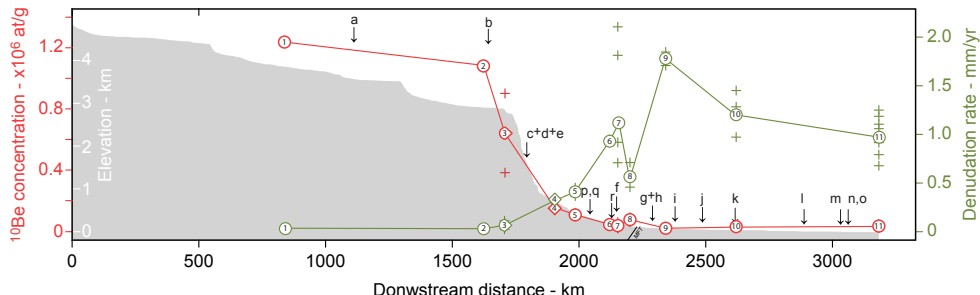

**Figure 3:** Downstream evolution of the [10]Be concentration (red) and corresponding denudation rates (green) of samples from the Tsangpo-Brahmaputra trunk stream. The elevation profile of the trunk stream channel is plotted in shaded grey. Down pointing arrows refer to the confluences with main Tsangpo-Brahmaputra tributaries. Open symbols are averages in the case where multiple measurements have been carried out and crosses represent individual measurements. Open circles are from this study 10 and open diamonds are samples from Finnegan et al. (2008).

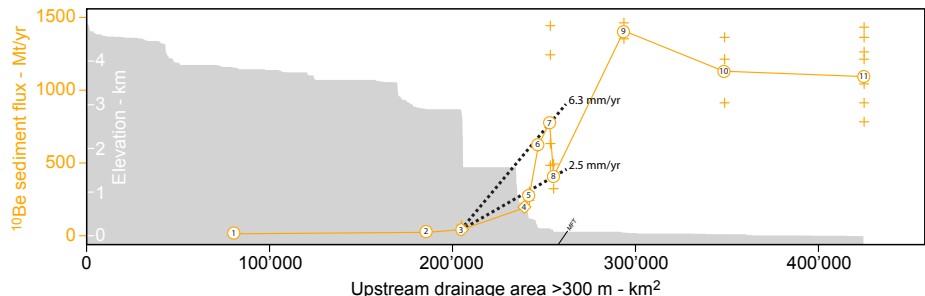

**Figure 4:** Evolution of the [10]Be-derived sediment flux as a function of the upstream drainage area above 300 m of elevation (in order to remove floodplain area storing sediments) from the Tsangpo-Brahmaputra trunk stream along with the channel elevation 15 profile in grey. The average denudation rate over a given reach of the river is given by the slope of the regression between sample points. Open symbols are averages in the case where multiple measurements have been carried out and crosses represent individual measurements. Open circles are from this study and open diamonds are samples from Finnegan et al. (2008)



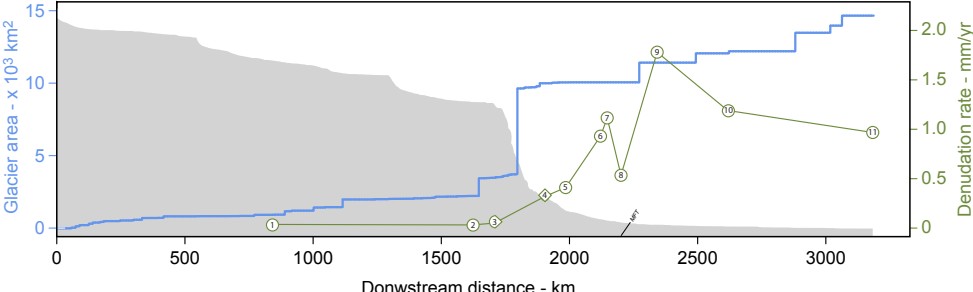

**Figure 5: Downstream evolution of the total drained glaciated area in blue (GLIMS database: Raup et al., 2007) of the Tsangpo-Brahmaputra. For comparison, the evolution of the trunk channel elevation is plotted in grey and the average [10]Be denudation rate in green.**

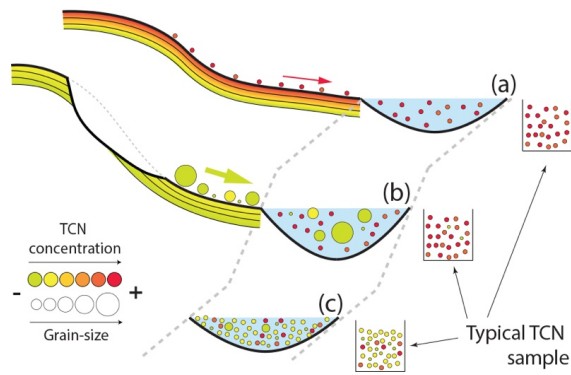

**Figure 6: Conceptual diagram illustrating the evolution of the sediment [10]Be signal in the Tsangpo-Brahmaputra. (a) high [10]Be concentration sediments are transferred to the channel in the slowly eroding upstream Tibetan part of the catchment. (b) The high concentration [10]Be flux is diluted by low-concentration sediment produced by landslides within the NBGPm region. However,**
10  **landslides mainly transfer coarse material to the channel so that the sediments sampled within the sand fraction that is typically used for [10]Be measurements is still dominated from material derived from the Tibetan plateau. (c) Further downstream, abrasion processes transfer sediment from the coarse-grained landslide material into the sand fraction so that the denudation signal of the NBGPm is only sampled downstream of the actual location of intense denudation.**





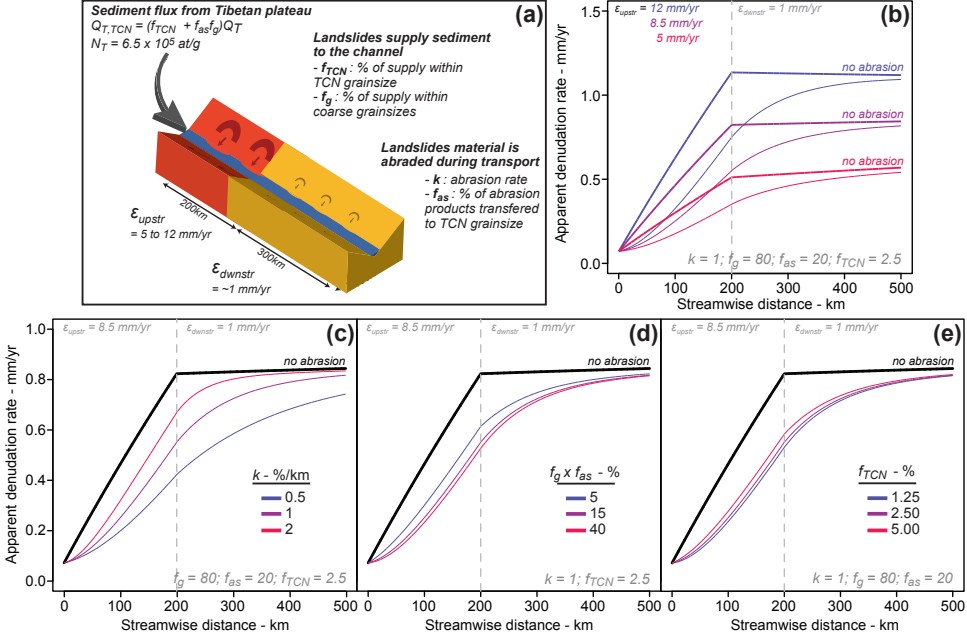

**Figure 7: (a)** – Schematic of the modeled river reach; main boundary conditions and process parameters used in the model. **(b)** to **(e)** are abrasion model results showing the apparent [10]Be denudation rate reconstructed from river sediments within the sand fraction in case of abrasion, compared to the no abrasion case (thick line). The sensitivity to changes in different parameters has been explored: **(b)** sensitivity to changes in absolute erosion rates over the modeled reach; **(c)** sensitivity to changes in the abrasion coefficient $k$; **(d)** sensitivity to changes in the product between the initial fraction of coarse material (>2mm) delivered to the channel $f_g$ and the fraction of this coarse material that is ultimately transferred to the TCN sampled and measured fraction $f_{as}$ and **(e)** sensitivity to changes in the initial fraction of sediment delivered by landslides directly within the TCN grain size fraction $f_{TCN}$. More information on the model setup and results can be found in the text.

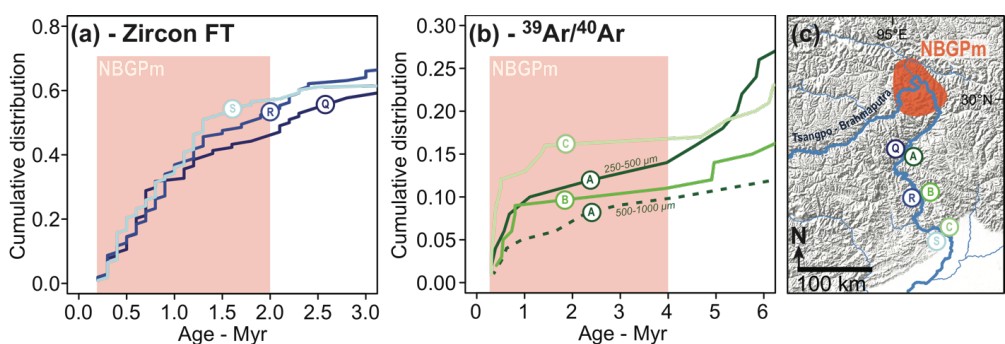

**Figure 8: (a)** Zircon Fission Track (Enkelmann et al., 2011) and **(b)** [39]Ar/[40]Ar (Lang et al., 2016) of sediment samples from the Tsangpo-Brahmaputra downstream to the NBGPm **(c)** (trimmed to the younger part of the sample age spectrum).





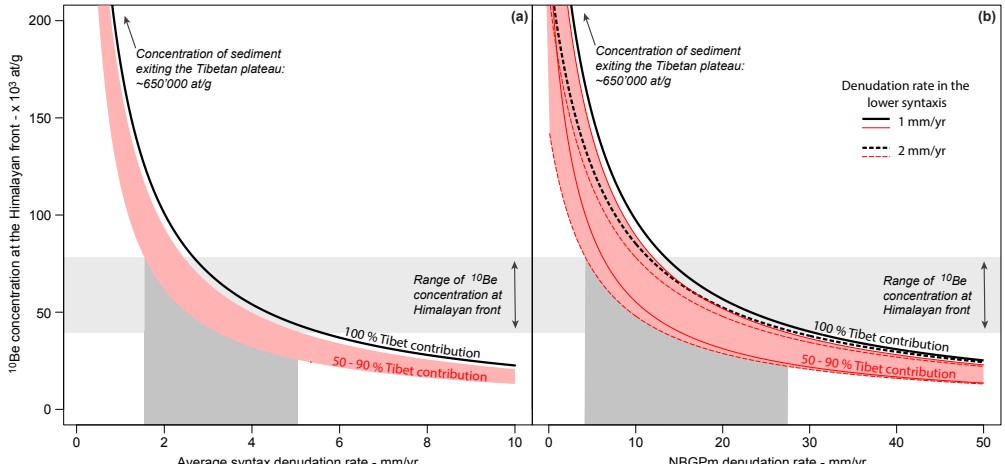

**Figure 9: $^{10}$Be concentration mixing model. (a) The concentration of sediments exported at the Himalayan front (pts. 7-8) is calculated by mixing sediments from the Tibetan plateau (pt. 3) with sediments produced along the entire syntaxis (pts. 3 to 7-8) for a range of denudation rates in the syntax. (b) the mixing model is calculated for variable denudation rates within the restricted**
5  **area of the NBGPm, mixing with sediments exported by the Tibetan plateau, tributaries within the syntaxis (pts. c, d, e) as well as for two estimates of denudation rates in the lower part of the syntaxis (1 and 2 mm/yr). The $^{10}$Be concentration modeled at the Himalayan front is then compared to the $^{10}$Be concentrations actually measured (pts. 7-8) in order to constrain the range of possible denudation rates in the entire syntaxis (a) and the NBGPm (b). The solid and dashed black curves are the modeled concentrations assuming that 100% of the sediment flux predicted by the TCN is exported by Tibetan part of the catchment.**
10  **However, as discussed in the text, sediment storage and lower quartz abundances imply fluxes that are likely only 50 to 90% of the TCN-derived flux. The model results for a reduced Tibetan contribution are shown by the red envelope and result in lower denudation rate estimates for the entire syntaxis (a. 1.8 to 5.1 mm/yr) and the NBGPm (b. 4 to 27 mm/yr).**

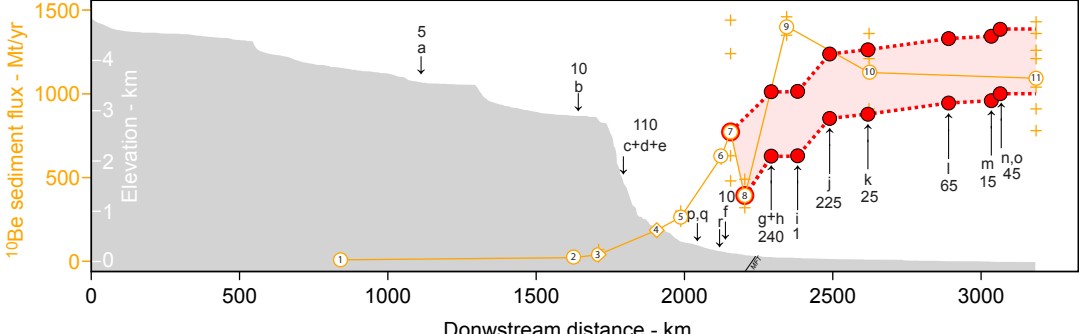

15  **Figure 10: Evolution of the $^{10}$Be-derived sediment flux as a function of the upstream drainage area above 300 m of elevation from the Tsangpo-Brahmaputra trunk stream along with the channel elevation profile in grey. The arrows show the addition of sediments from tributaries along with their $^{10}$Be-derived sediment addition in Mt/yr. The red curves show the evolution of the total sediment flux that is obtained from adding tributary sediment fluxes to the Tsangpo-Brahmaputra sediment flux that exists the Himalayan range (with a flux of 770 Mt/yr at pt. 7 or 390 Mt/yr at pt. 8).**

