# Peer review of "10Be systematics in the Tsangpo-Brahmaputra catchment: the cosmogenic nuclide legacy of the eastern Himalayan syntaxis"

_Earth Surface Dynamics, 2017_

## Referee Comment (RC1) · Anonymous Referee #1 · 4 May 2017

General comments

The manuscript is unusually thorough and generally presents well-considered arguments. The dataset is impressive and the authors should be commended for their painstaking incorporation of previously published data. While I am not convinced that the their grain-abrasion argument is the most parsimonious explanation for their observations – it is certainty a contributing factor and their thorough discussion of the model is a valuable contribution to the community. Overall, nicely done.

Specific comments

While grain abrasion may be important for the interpretation of your CRN dataset

(amongst many other factors) it is not clear to me that it is similarly important for other single-grain detrital studies (as suggested in 27, first paragraph on p. 12, p. 16 line 15). Your analyses of published detrital mineral cooling ages is interesting, but not definitive (e.g. you do not demonstrate a significance difference between these samples using a statistical metric that accounts for sample size) and does not necessarily demonstrate that "care should be taken before relating detrital signals to spatial patterns of denudation within the NBGPm" (p. 12 13-14). I'm not saying we should be careless, but it is not clear that your data allow you to make this claim. Please clarify that, for the interpretation of relatively large aliquot quartz data, you prefer this explanation, and you *speculate* that this may also be true for single-grain analyses of other minerals.

The increased [10Be] between samples at site 7 and 8 is probably due to recycling of quartz rich, possibly high [10Be] terrace and Siwalik (Upper and Middle Siwalik here) sediments and does not reflect any sort of "inherent variability of the sediment transport system" (p. 7, 24). That variability should be reflected in your multi-year sampling, or sampling of the same place by different researchers. You should change the manuscript to reflect, or at least consider this interpretation. See also p. 8, 40 on.

I don't find your argument on p. 8, lines 38-39 convincing since the sediment flux varies dramatically throughout the catchment area. What really matters is not actually the catchment area, but the ratio of the landslide volume to the catchment-averaged sediment flux. While it is true that the catchment area downstream of the NBGPm is very large, the sediment flux from most of that upstream area is very low (∼10 Mt/yr) – so does it matter? Something to think about.

Landsliding is clearly the dominant process transporting sediment to channels in this landscape (see Larsen's work, primarily), but is it true that these are all deep-seated bedrock landslides that produce coarse debris? These hillslopes are remarkably soil mantled. It would be worth looking at the area/frequency distribution of landslides here to see if they are actually just shallow soil slips that deliver fine sediment, not coarse debris. Something else to think about.

Technical corrections/errors/etc by line

The manuscript is clear, but a little wordy and sometimes the phrasing is awkward. I've highlighted a couple examples, but the text would benefit from a brief read-through by a couple colleagues.

p. 1 31. remove "other to" 35. Awkward phrasing, consider "As a consequence..., the Himalayan range is the ... ocean." 38. Awkward, "because of variable physiography, geomorphological processes and climate." 40. A convenient, but not correct definition of syntaxes, consult geologic dictionary or rephrase to sound less like a definition, e.g. "Interactions between tectonic and surface processes are most likely observed in tectonic syntaxes."

p. 2 25. "simply reading...." does not need quotation marks.

p. 3 21. See work by AK Jain, Jain and Thakur (mid-late 70s in Himalayan Geology) for first mapping the region. SK Acharyya for regional compilation. These are better citations. 26. Arunachal Pradesh is the name of the state, not just "NE."

p. 8. 1. Salvi et al. is published. 5. Has this actually been demonstrated by Larsen and Montgomery? I'm not sure that any landslide density estimates are published along the Siang. Consider rephrasing.

p. 11 35. odds with 37. in detail the

p. 12 34. need to be

p. 13 1. if you have mapped the sand/gravel transition, it would be valuable to plot this in your figures. 25. not clear what you are saying after "but thermochronological data.... " End the sentence after the parenthetical.

p. 14 18. Unclear where these number (50 and 90%) come from, please explain/rephrase.

**ESurfD**

Interactive
comment

---

## Referee Comment (RC2) · Anonymous Referee #2 · 6 May 2017

Lupker and colleagues show 10Be concentrations in river sand along the Tsangpo-Brahmaputra river. This river crosses the Namche Barwa-Gyala Peri massif, uplifting and eroding at a higher rate than upstream and downstream. Converting the river-ine 10Be concentrations in mean denudation rates of catchment above each sampling point, they observe that the peak of denudation rate is shifted almost 500 km down-stream from the uplift hotspot. Without knowing that the Namche Barwa-Gyala Peri massif was an area of high denudation rate, the 10Be-derived denudation rates would have led to a wrong zonation of the denudation rate in the eastern Himalayan syntaxis. Lupker and colleagues argue that this mismatch arises from a delay imposed by the abrasion of coarse bedload sediment. Indeed, sediment generated in the hillslopes of

the uplift hotspot may include a significant fraction of coarse material > 1mm. Yet, the 10Be concentration is measured in the [0.5-1] mm fraction. A long transport distance is required for the coarse material to be crushed into the [0.5-1] mm fraction. Using a simple model based on estimates of abrasion rate and mass balance of 10Be fluxes, Lupker and colleagues show that their interpretation is likely.

These data have a significant impact concerning the determination of catchment-scale denudation rates derived from cosmogenic nuclides concentrations. They show that the dilution process, first proposed and discussed by Belmont et al. (2007), can strongly alter the mapping of denudation rate in catchments. In turns, this study opens perspectives of using riverine cosmogenic nuclides to quantify abrasion rates in mountain rivers. The paper is well written and the data and model support the conclusion. That said, I have several concerns I would like to be discussed.

1- In Figure 3, the mean denudation peak (circles) occurs at point 9, almost 500 km downstream the high uplift rate area. Nevertheless, there is a significant variability in 10Be concentrations between individual samples at the same site. In particular, at point 7, two individual measurements (crosses) show high denudation rates, as high as the peak at point 9. The average (circle) at point 7 is much lower because there are 2 individual samples corresponding to low denudation (why is not the mean at equal distance between the two high and two low values ?). How do you interpret this variability at point 7 and the high denudation rate (low 10Be concentration) points ?

2- The results are presented on the form of upstream mean catchment rates. Showing the mean denudation rates of nested catchments, in particular between points 3 and 4, would help see the discrepancy between the high uplift rate area and its low mean denudation.

3- The model is simple and its outcomes seem conclusive. However, how does the predicted denudation pattern change if the true drainage areas are taken into account instead of assimilating the catchment to a rectangle ? I think that the model outcomes,

presented as a synthetic case study, is demonstrative. Nevertheless, this model (the best fit one for example) would deserve to be compared to the data along the river profile.

4- I was very surprised that the model prediction does not depend significantly on the product fg x fas which determines the amount of coarse material provided by the high uplift area to the river sediment downstream. It is very striking that adding 5% or 40% of this material to the [0.5-1] mm fraction leads almost to the same result (Figure 7d). Is this result robust for different abrasion rates k ? How is it possible that adding to the [0.5-1] mm fraction only 5% of the coarse material delivered by the high uplift area has such an impact ? On Figure 7c, I presume that the legend is wrong, namely that the downstream actually increases with the abrasion rate k, isn't it ?

5- This study demonstrates that the sand fraction provides a low estimate of the denudation rates in rapidly eroding area. This echoes suggestions in previous studies (e.g. Belmont et al., 2007; Aguilar et al., 2014; Carretier et al, Quat. Geoch., 2015), and demonstrates it clearly here. The following question may be beyond the scope of this paper, but broadly speaking, should we sample and crush together all the bedload grain sizes (including cobbles) to get a better estimate of the denudation rate ?

Specific comment by lines.

Page 2 line 35 You may want to cite Belmont et al. (2007) who discussed the effect of pebble abrasion and the dilution effect.

Belmont, P., Pazzaglia, F., Gosse, J., 2007. Cosmogenic 10Be as a tracer for hillslope and channel sediment dynamics in the Clearwater River, western Washington State. Earth Planet. Sci. Lett. 264, 123e135.

Page 5 line 18 The latitude Stone (2000) scaling scheme for muons is obsolete. The relative contribution of muons and neutrons varies with elevation. Could you specify if you took this into account ?

Page 5 line 34 The absence of incision could mean that there is some addition of floodplain sediment and 10Be concentration to the river bedload by lateral erosion. I guess this would not change your results, but you may discuss this.

Page 9 line 7 Please justify why this recent sediment input is unlikely to dominate the sediment budget.

Page 10 line 2 Do you consider that sand is mainly in suspension ?

Page 11 Please justify the range of tested abrasion parameters.

Page 11 line 31-32 I do not understand this sentence.

Page 11 line 34. I think you should not too much worry about the absence of systematic correlation between grain size and 10Be concentration in this case. This comparison concerns very fine fractions, with a lot of different possible origins. It is possible that the by-products of abrasion that affect the riverine 10Be signal downstream derives from the abrasion of large pebbles and cobbles. Obviously you can not prove this because you do not show 10Be concentration of this fraction, but Carretier et al. (Quaternary Geochronology, 2015) for example, showed a systematic low 10Be concentration in 10 cm pebbles in several catchments along the western Andes, whereas the 10Be concentration of 2 cm gravel is either lower or higher than the sand fraction. In an Himalayan catchment, Puchol et al (2014) showed lower 10Be concentrations in [0.47-4] cm pebbles. This remains to be proven, but if rapidly eroding areas provide a large proportion of sediment as large pebbles and cobbles, systematically less concentrated in 10Be, then this may be the abrasion of that coarse material that controls the dilution process downstream, and not that of the smaller fraction. If true, knowing the size distribution of material eroded from the hillslopes would be essential to quantitatively interpret the 10Be riverine signal in terms of abrasion rate. Again, it is clear that you cannot provide more information about this debate with your data, but it may be worth to think about it.

Page 16 line 30 takinG

Figure 3: localize the Namche Barwa-Gyala Peri massif on that profile and the others.

Figure 7c: wrong legend.

Figure 10: I do not fully understand what are the two red lines, the red circle and the red zone between the two lines. The caption indicates incorrectly that the x axis is the upstream drainage area whereas it is the downstream distance. "exists" → exits.

---

## Author Comment (AC1) · 13 Jun 2017

Answer to reviewer comments and annotated manuscript are in the attached pdf.

Please also note the supplement to this comment:
http://www.earth-surf-dynam-discuss.net/esurf-2017-18/esurf-2017-18-AC1-supplement.pdf

---

## Author Response (AR1)

**Answer to reviewers for manuscript esurf-2017-18:**
*[10]Be systematics in the Tsangpo-Brahmaputra catchment: the cosmogenic nuclide legacy of the eastern Himalayan syntaxis*

Reviewer's comment
*Answer to comment*
Modification in the manuscript

**Anonymous Referee #1**

General comments
The manuscript is unusually thorough and generally presents well-considered arguments. The dataset is impressive and the authors should be commended for their painstaking incorporation of previously published data. While I am not convinced that the their grain abrasion argument is the most parsimonious explanation for their observations – it is certainty a contributing factor and their thorough discussion of the model is a valuable contribution to the community. Overall, nicely done.
*We'd like to thank reviewer for his constructive comments. The abrasion argument is only introduced after reaching the conclusion that other more "classical" perturbations could not explain the observed trend.*

Specific comments
While grain abrasion may be important for the interpretation of your CRN dataset (amongst many other factors) it is not clear to me that it is similarly important for other single-grain detrital studies (as suggested in 27, first paragraph on p. 12, p. 16 line 15). Your analyses of published detrital mineral cooling ages is interesting, but not definitive (e.g. you do not demonstrate a significance difference between these samples using a statistical metric that accounts for sample size) and does not necessarily demonstrate that "care should be taken before relating detrital signals to spatial patterns of denudation within the NBGPm" (p. 12 13-14). I'm not saying we should be careless, but it is not clear that your data allow you to make this claim. Please clarify that, for the interpretation of relatively large aliquot quartz data, you prefer this explanation, and you *speculate* that this may also be true for single-grain analyses of other minerals.
*This part has been rephrased so as to moderate our statement on the effect of abrasion on thermochronological data in the area of the eastern Himalaya syntaxis as suggested by the reviewer. The two-sample Kolmorgorov-Smirnov test does not suggest that the NBGPm signal is diluted downstream as would be expected if a significant amount of sediments from downstream of the NBGPm were added to the mainstream flux. The downstream (pt. C) $^{39}Ar/^{40}Ar$ sample contains significantly (at the p = 0.1 level) more young age grains than samples A and B. The populations are however not significantly different for the three ZFT samples. The literature data therefore provide some evidence that the NBGPm signal increases downstream of the NBGPm but more samples would be needed to ascertain this suggestion. However, if the micas, zircons and quartz content of the eroded source rocks are constant and if the suggestion that abrasion is an important mechanism is true, we do not see any mechanism that would explain how this abrasion effect could be observed for quartz but not for other minerals. For any detrital signal to be picked-up in the sand fraction, abrasion processes need to transfer the eroded mass from large clasts to the sand fraction.*
p.12 l.16: Abrasion effects in the Tsangpo-Brahmaputra are unlikely to be restricted to detrital TCN studies. Other detrital proxies such as thermochonological ages of sediment grains could be affected similarly, i.e. the young cooling ages found in the NBGPm (e.g. Zeitler et al., 2014) could be transferred to the typically studied sand fraction of the sediment load further downstream only after coarse landslide material is abraded. The population of Zircon fission track (FT) ages (Enkelmann et al., 2011) and $^{39}Ar/^{40}Ar$ ages (Lang et al., 2016) were measured in three Tsangpo-Brahmaputra river sediment samples from downstream of the NBGPm to the Himalayan front (Fig. 8). In both cases, the relative proportion of the youngest grains (< 2 Myr for Zircon Ft and < 4 Myr for $^{39}Ar/^{40}Ar$) does not significantly decrease downstream of the NBGPm (based on a two-sample, one sided, Kolmogorov-Smirnov test at the p = 0.1 level) as would be expected if the NBGPm signal were diluted by sediments from lower exhuming areas. While the Zircon Ft populations of the three samples are not significantly different, the $^{39}Ar/^{40}Ar$ data shows that the proportion of young grains increases from A to C (p = 0.1). This later observation is expected if the NBGPm rapid exhumation signal is transferred to the sand fraction only after abrasion of the coarse NBGPm landslide material. More samples are obviously needed to further test this hypothesis but we speculate that similar abrasion should be considered in actively eroding areas for a range of detrital sediment provenance and denudation tracers.

The increased [10Be] between samples at site 7 and 8 is probably due to recycling of quartz rich, possibly high [10Be] terrace and Siwalik (Upper and Middle Siwalik here) sediments and does not reflect any sort of "inherent variability of the sediment transport system" (p. 7, 24). That variability should be reflected in your multi-year sampling, or sampling of the same place by different researchers. You should change the manuscript to reflect, or at least consider this interpretation. See also p. 8, 40 on.

*We added the Siwalik recycling hypothesis but it is not our favoured one. The two samples have been taken two years apart, once from the river right bank and once from the left bank, close to the shore during low flow. Both have been taken from upstream of the most obvious derived Siwalik inputs (specially a large Siwalik input just North of Pasighat). In addition of these sampling location considerations, it would take a very large Siwalik input to significantly perturb the ca. 500 Mt of sediments per year transported by the Tsangpo-Brahmaputra in Pasighat during the two sampling dates.*

**p.7 l.38:** Discrepancies between consecutive upstream and downstream estimates, and especially between pts. 7 and 8, suggest local perturbations of the sediment transport system. The cause of these perturbations remains difficult to explain. Recycling of Siwalik sediments could explain such a trend but the Siwalik-drained area is extremely small (<100 km$^2$, Acharyya et al. 2007) so that a steady state Siwalik contribution would be negligible. In addition, our two samples have been sampled two years apart, from the two opposing banks which should avoid the sampling of sudden, local and stochastic Siwalik inputs.

I don't find your argument on p. 8, lines 38-39 convincing since the sediment flux varies dramatically throughout the catchment area. What really matters is not actually the catchment area, but the ratio of the landslide volume to the catchment-averaged sediment flux. While it is true that the catchment area downstream of the NBGPm is very large, the sediment flux from most of that upstream area is very low (10 Mt/yr) – so does it matter? Something to think about.

*The upstream area has a very low denudation rate and covers a large area so that exported sediment fluxes are low (as pointed at by the reviewer). Landslide frequency is therefore presumably much lower than in the NBGPm part of the catchment so that the TCN signal at the upstream entrance of the NBGPm should not be affected in a significant way by landslide biases. The potential biases only arise for sediments in the NBGPm area, where landsliding is frequent. There, we agree with the reviewer that the effects of landslides cannot only be evaluated with respect to the upstream area given that the denudation rate is not uniform. The Niemi et al., (2005) and Yanites et al. (2009) models do not cover cases with non-uniform denudation across the catchment and, to our knowledge, no systematic study has been carried out in this direction. The effect of landslides will be most important in the upstream area, close to the NBGPm when the overall sediment flux is still relatively small. There, large landslides will have an important impact on the TCN signal since the upstream sediment flux is low and the difference in the TCN concentration between landslide material and Tibet-derived river sediments is important. However, the riverine sediment flux increasing in the downstream direction, the effect of landslides will also decrease (the ratio of landslide volume to catchment-averaged sediment flux decreases) so that samples downstream of the NBGPm should be less affected. It is nevertheless still a possibility and we have modified the text accordingly:*

**p.9 l.2:** Modelling studies suggest that this effect is more pronounced for small catchments of a few tens of km2 or less and the magnitude of the perturbation depends on the ratio of landslide volume to catchment-averaged sediment flux (Niemi et al., 2005; Yanites et al., 2009). The catchments drained by the Tsangpo-Brahmaputra downstream of the NBGPm exceed 200000 km$^2$ but these are characterized by a very heterogeneous denudation rate. The riverine TCN signal will be prone to perturbation in the upstream reaches of the syntaxis region, where the upstream sediment flux is still low and the difference in the 10Be concentration between the sediments carried by the river and landslide material is high. Further downstream of the NBGPm, the effect of landslides should be less important because of the rapidly increasing river sediment flux and decreasing [10]Be concentration and we therefore expect that regular and diffuse landsliding is likely not a major source of perturbation for CWDRs downstream of the NBGPm.

Landsliding is clearly the dominant process transporting sediment to channels in this landscape (see Larsen's work, primarily), but is it true that these are all deep-seated bedrock landslides that produce coarse debris? These hillslopes are remarkably soil mantled. It would be worth looking at the area/frequency distribution of landslides here to see if they are actually just shallow soil slips that deliver fine sediment, not coarse debris. Something else to think about.

*This is an interesting remark that we did not explore in the manuscript. As suggested by the reviewer, it is possible to calculate the proportion of material that is derived from bedrock landslide based the NBGPm landslide inventory and statics published by Larsen and Montgomery (2012). In that contribution, the authors assume a threshold area of $10^5$ $m^2$ between soil and bedrock dominated landslides and integrating the area-volume scaling and the frequency distribution above this threshold area suggests that bedrock derived landslides contribute to about 95% of the total volume (cf. figure).*

[Figure]

*However, a number of factors could potentially complicate the picture: first and foremost, there is very little to no direct observation of the type of landslides or hillslope cover in the NBGPm area. The threshold value of $10^5$ $m^2$ suggested by Larsen and Montgomery to separate bedrock from soil mantled landslides is derived from a global compilation and may therefore not adequately represent the NBGPm. Our own field experience across the Himalayan range suggests that hillslope soil generally develops in colluvial material, previous landslide deposits or highly disrupted regolith and that soils typically contain a large proportion of coarse material that can be transferred to the river during landsliding. The studies by Puchol et al., (2014) and Gallo and Lavé (2014) of a landslide in the Nepal Himalaya also suggest that for an area of 0.5 $km^2$ (the used threshold), the proportion of coarse debris dominates the mobilised sediment flux. Furthermore, our abrasion model depends on the proportion of material above 2mm that can be abraded during transport and is therefore not dependent on the precise grain-size distribution of coarse landslide material. While we appreciate this remark, we argue that in the absence of more data on the types of soil cover and landslides in the NE-Himalaya we cannot quantitatively use the landslide statistics to refine our abrasion model. The landslide inventory furthermore suggests that shallow soil landslides only represent a limited overall volume.*

Technical corrections/errors/etc by line
The manuscript is clear, but a little wordy and sometimes the phrasing is awkward. I've highlighted a couple examples, but the text would benefit from a brief read-through by a couple colleagues.
*The manuscript has been read by the co-authors and the most obvious mistakes corrected.*

p. 1 31. remove "other to" 35. Awkward phrasing, consider "As a consequence, … the Himalayan range is the … ocean."
*Done*

38. Awkward, "because of variable physiography, geomorphological processes and climate."
*Rephrased*

40. A convenient, but not correct definition of syntaxes, consult geologic dictionary or rephrase to sound less like a definition, e.g. "Interactions between tectonic and surface processes are most likely observed in tectonic syntaxes."

*Rephrased*

p. 2 25. "simply reading…." does not need quotation marks.
*Removed*

p. 3 21. See work by AK Jain, Jain and Thakur (mid-late 70s in Himalayan Geology) for first mapping the region. SK Acharyya for regional compilation. These are better citations.
*Two of the suggested citations were added instead of Bhat et al.:*
Jain, A. K. and Thakur, V. C.: Abor Volcanics of Arunachal Himalaya, Journal of the Geological Society of India, 19, 335-349, 1978.
Acharyya, S. K.: Evolution of the Himalayan Paleogene foreland basin, influence of its litho-packet on the formation of thrust-related domes and windows in the Eastern Himalayas – A review, Journal of Asian Earth Sciences, 31, 1-17, 2007.

26. Arunachal Pradesh is the name of the state, not just "NE."
*Corrected*

p. 8. 1. Salvi et al. is published.
*Corrected*
Salvi, D., Mathew, G., and Kohn, B.: Rapid exhumation of the upper Siang Valley, Arunachal Himalaya since the Pliocene, Geomorphology, 284, 238-249, 2017.

5. Has this actually been demonstrated by Larsen and Montgomery? I'm not sure that any landslide density estimates are published along the Siang. Consider rephrasing.
*True, the work by Larsen and Montgomery (2012) only shows decreasing landsliding rates downstream of the NBGPm but along the upper Siang. There is no published landslide rate data along the lower Siang.*
**p.8 l.9:** The recent erosional activity displays a similar pattern with a maximum landslide density in the NBGPm around the above-mentioned confluence, and which significantly decrease further downstream, along the upper Siang (Larsen and Montgomery, 2012).

p. 11 35. odds with
*Corrected*

37. in detail the
*Corrected*

p. 12 34. need to be
*Corrected*

p. 13 1. if you have mapped the sand/gravel transition, it would be valuable to plot this in your figures.
*Given the scale of the plots, the proximity between the GST and the MFT would make it confusing on the figures.*

25. not clear what you are saying after "but thermochronological data…. " End the sentence after the parenthetical.
*Corrected*
**p.13 l.37:** Thermochronological data also suggest exhumation rates significantly lower than the NBGPm (Salvi et al., 2017) suggesting overall low denudation rates.

p. 14 18. Unclear where these number (50 and 90%) come from, please explain/ rephrase.
*The previous paragraph discusses the upper and lower bounds of the reduction in sediment flux exported by the Tibetan Plateau part of the catchment. These estimates are based on qualitative or poorly documented datasets in the literature. We therefore clearly state that our estimate of 50 to 90 % is a speculation that is broad enough to provide a conservative estimate and uncertainty range. We added a sentence to clarify that this reduction needs to better quantified.*
**p.14 l.28:** We conservatively speculate here, that the actual flux exported by the Tibetan plateau part of the catchment ranges between 50 and 90 % of the flux that would be calculated by the TCN. The exact magnitude of this sediment flux reduction remains to be quantified.

**Anonymous Referee #2**

Lupker and colleagues show 10Be concentrations in river sand along the Tsangpo- Brahmaputra river. This river crosses the Namche Barwa-Gyala Peri massif, uplifting and eroding at a higher rate than upstream and downstream. Converting the riverine 10Be concentrations in mean denudation rates of catchment above each sampling point, they observe that the peak of denudation rate is shifted almost 500 km downstream from the uplift hotspot. Without knowing that the Namche Barwa-Gyala Peri massif was an area of high denudation rate, the 10Be-derived denudation rates would have led to a wrong zonation of the denudation rate in the eastern Himalayan syntaxis. Lupker and colleagues argue that this mismatch arises from a delay imposed by the abrasion of coarse bedload sediment. Indeed, sediment generated in the hillslopes of the uplift hotspot may include a significant fraction of coarse material > 1mm. Yet, the 10Be concentration is measured in the [0.5-1] mm fraction. A long transport distance is required for the coarse material to be crushed into the [0.5-1] mm fraction. Using a simple model based on estimates of abrasion rate and mass balance of 10Be fluxes, Lupker and colleagues show that their interpretation is likely.

These data have a significant impact concerning the determination of catchment-scale denudation rates derived from cosmogenic nuclides concentrations. They show that the dilution process, first proposed and discussed by Belmont et al. (2007), can strongly alter the mapping of denudation rate in catchments. In turns, this study opens perspectives of using riverine cosmogenic nuclides to quantify abrasion rates in mountain rivers. The paper is well written and the data and model support the conclusion.

That said, I have several concerns I would like to be discussed.

1- In Figure 3, the mean denudation peak (circles) occurs at point 9, almost 500 km downstream the high uplift rate area. Nevertheless, there is a significant variability in $^{10}$Be concentrations between individual samples at the same site. In particular, at point 7, two individual measurements (crosses) show high denudation rates, as high as the peak at point 9. The average (circle) at point 7 is much lower because there are 2 individual samples corresponding to low denudation (why is not the mean at equal distance between the two high and two low values ?). How do you interpret this variability at point 7 and the high denudation rate (low $^{10}$Be concentration) points ?

*First, the average denudation rate (circle) is not the arithmetic mean of the denudation rates (crosses) because we used the average concentration of the samples and converted that to a denudation rate (instead of the average of the calculated denudation rates). Since denudation rates and concentration are not linearly related this may lead to significant differences in the estimated mean denudation rate. This has been more clearly stated in the caption of Fig. 3 & 4:*

Open symbols are based on the average $^{10}$Be concentration of all samples when multiple samples have been analyzed…

*The variability at pt. 7 is observed between two grain-size fractions (respectively 250-500 and 500-1000 µm) of a same sediment sample. It is the only sample we have that shows a significant difference between two different grain sizes (see Fig. 2). To ascertain this difference, these two samples have been duplicated (duplicate of the whole quartz purification, chemical separation and AMS measurement) and unfortunately one of the two duplicates did not reproduce very well (500-1000 µm). Since there is no argument for discarding one of these measurements or grain-sizes we chose to present all the data in the graphs and figures and take the average as the most representative value. The reasons behind the difference in concentration among these two samples remains unclear. Since it is the only sample with such a behavior we simply suggest it being a local exception resulting from an odd mixing effect between two different sediments sources with different grain sizes. No significant amount of natural $^{9}$Be that could have affected the final $^{10}$Be concentration estimate was found in these samples either. We added the following to the manuscript:*

**p.6 l.31:** The difference in concentration for these two grain size classes is robust since the quartz purification procedure, chemical separation and AMS measurement has been duplicated but in the absence of any obvious explanation we suggest that this difference might be due to the mixing of sediment from different sources with different grain-sizes and $^{10}$Be concentrations.

2- The results are presented on the form of upstream mean catchment rates. Showing the mean denudation rates of nested catchments, in particular between points 3 and 4, would help see the discrepancy between the high uplift rate area and its low mean denudation.

*We added some of the denudation rates between the nested catchments directly in the text. We however don't want to overemphasize these estimates as the difference between subsequent measurement points is very sensitive to the variability of the signal and would for instance be negative between pt.7 and pt.8. It is probably much more meaningful to evaluate average denudation rates between nested catchments over a number of sampling points, which has also been done.*

**p.7 l.35:** The apparent denudation rate of the NBGPm area, calculated between pts. 3 and 4, equals ca. 1.6 mm/yr. The highest denudation rates however occur further downstream of the NBGPm and peak between pts. 5 and 6 with an apparent denudation rate of 26 mm/yr and are overall higher than in the NBGPm between pts. 4 and 7 (15 mm/yr). Discrepancies between consecutive upstream and downstream estimates, and especially between pts. 7 and 8, suggest local perturbations of the sediment transport system. The cause of these perturbations remains difficult to explain. Recycling of Siwalik sediments could explain such a trend but the Siwalik-drained area is extremely small (<100 km$^2$, Acharyya et al. 2007) so that a steady state Siwalik contribution would be negligible. In addition, our two samples have been sampled two years apart, from the two opposing banks which should avoid the sampling of sudden, local and stochastic Siwalik inputs.

3- The model is simple and its outcomes seem conclusive. However, how does the predicted denudation pattern change if the true drainage areas are taken into account instead of assimilating the catchment to a rectangle ? I think that the model outcomes, presented as a synthetic case study, is demonstrative. Nevertheless, this model (the best fit one for example) would deserve to be compared to the data along the river profile.

*This is a tricky question to answer. Our simple model is aimed at proving that abrasion processes of coarse landslide material can be responsible for the downstream lag of the TCN response measure in the trunk river. To show this, we model the reach with a simplified river section. Eventhough, our section is simplified, it still satisfies the area, sediment flux and production rate parameters of the actual eastern Himalayan syntaxis. Directly comparing the intermediate measurements with the model, would require the model to spatially resolve confluences (for instance with the Yigong and Parlung) and hence would require us to implement a spatial distribution of landslides. This would slowly lead towards the implementation of TCN systematics in a landscape evolution model which would certainly be interesting but is beyond the scope of this work. Moreover, we believe that the shape of the catchment is also a relatively minor source of uncertainty compared to other parameters such as changes in abrasion rate depending on lithology or transport velocity (Attal and Lavé, 2009) or the quartz abundance of the source rock to name a few and that overall only a first order model is adequate. We nevertheless argue that the most relevant parameter predicted by our model is the downstream delay or lag of the TCN response to intense erosion and landsliding and not the rate at which the signal is transferred to the sand fraction. In that respect, we believe that our simplified model still provides a useful estimate.*

**p.11 l.40:** However, our model only takes into account abrasion along the main-stream of the Tsangpo-Brahmaputra and does not take into account the abrasion of landslide material that may have occurred in smaller tributaries before reaching the trunk stream. Our model may therefore slightly overestimate the role of abrasion in generating the observed downstream lag in the denudation signal. Testing the limits of this model and directly comparing it to the data in the eastern syntaxis would require the spatial distribution of landslides and drainage area to be modeled explicitly, which is beyond the scope of this contribution.

4- I was very surprised that the model prediction does not depend significantly on the product fg x fas which determines the amount of coarse material provided by the high uplift area to the river sediment downstream. It is very striking that adding 5% or 40% of this material to the [0.5-1] mm fraction leads almost to the same result (Figure 7d). Is this result robust for different abrasion rates k ? How is it possible that adding to the [0.5-1] mm fraction only 5% of the coarse material delivered by the high uplift area has such an impact ? On Figure 7c, I presume that the legend is wrong, namely that the downstream actually increases with the abrasion rate k, isn't it ?

*We verified our model and the results are robust for different values of k (see attached figure).*

[Figure]

We see two reasons for the observed model response:

- *The first reason for $f_g \times f_{as}$ to have only a limited impact on the final model outcome is that these parameters also apply to the sediment flux exported by the Tibetan plateau part of the catchment. The total amount of sediments that will ultimately be exported in the TCN fraction is set by $Q * (f_{TCN} + (f_g \times f_{as}))$, where $Q$ is total sediment flux exported by a reach. In our model, similar abrasion processes are also expected to occur in the upstream Tibetan part of the reach. That means that the upstream Tibetan part only exports a fraction of its sediment load within the TCN grain size (this fraction being $f_{TCN} + (f_g \times f_{as})$). The dilution in the eastern syntaxis is then also a function of the same abrasion coefficients. In other words, for a low $f_g \times f_{as}$, the amount of sediment transferred from landslides to the TCN fraction in the NBGPm is low but it also means that the fraction of sediment in the TCN fraction coming from the Tibetan plateau upstream of the NBGPm is low too. For high $f_g \times f_{as}$ values, the high amounts of landslide material transferred to the TCN fraction is counter balanced by the high amount of TCN-sized sediment exported upstream. In the end these parameters have therefore a limited impact on the modelled signal. This was already stated in the manuscript:*

  **p.11 l.16:** The fraction of this flux within the TCN grain-size assumes that all coarse material has been abraded: $$Q_{T,TCN} = (f_{TCN} + f_g f_{as})Q_T.$$

- *A second reason is that in our model scenarios, the ratios $f_g \times f_a / f_{TCN}$ is rather high, meaning that the system is dominated by abrasion products ($f_g \times f_a$), rather than by direct sediment output from the landslides within the TCN fraction ($f_{TCN}$). In such a case the sensitivity to $f_g \times f_a$ is limited. For systems with low $f_g \times f_a / f_{TCN}$ ratios this sensitivity is present (as shown by the offset between the no-abrasion case and our modelled cases), but these scenarios have not been explored as they would lead to unrealistic $f_g$, $f_a$ and $f_{TCN}$ values.*

5- This study demonstrates that the sand fraction provides a low estimate of the denudation rates in rapidly eroding area. This echoes suggestions in previous studies (e.g. Belmont et al., 2007; Aguilar et al., 2014; Carretier et al, Quat. Geoch., 2015), and demonstrates it clearly here. The following question may be beyond the scope of this paper, but broadly speaking, should we sample and crush together all the bedload grain sizes (including cobbles) to get a better estimate of the denudation rate ?

*Yes, that is right, the model runs show that within the syntaxis the model output under predicts the true denudation rate. This however requires the presence of an upstream, high $^{10}$Be concentration sediment flux, as the abrasion model alone would not generate such a bias. This bias disappears further downstream when abrasion have abraded all the coarse material. In some cases this may however not be the case, for instance if coarse material is sequestered in fans or valley fillings. Crushing all the grain-sizes would be theoretically be an option but this seems first impractical and would require the coarse vs sand to be mixed in the proportion of their flux which would be very difficult to constrain. We would rather emphasize a more systematic measurement of TCN across grain-size fractions to recognize potential biases.*

Specific comment by lines.
Page 2 line 35 You may want to cite Belmont et al. (2007) who discussed the effect of pebble abrasion and the dilution effect. Belmont, P., Pazzaglia, F., Gosse, J., 2007. Cosmogenic 10Be as a tracer for hillslope and channel sediment dynamics in the Clearwater River, western Washington State. Earth Planet. Sci. Lett. 264, 123e135.
*The reference was added:*

**p.2 l.32:** Different erosion processes also affect the final grain-size of fluvially exported sediment, which may in turn lead to a grain-size dependence of TCN signals that needs to be taken into account in order to derive robust CWDR (Clapp et al., 2002; Belmont et al., 2007; Aguilar et al., 2014; Puchol et al., 2014).

Page 5 line 18 The latitude Stone (2000) scaling scheme for muons is obsolete. The relative contribution of muons and neutrons varies with elevation. Could you specify if you took this into account ?

*Muon production was scaled with altitude and latitude. The Stone (2000) scaling does take into account a separate altitudinal and latitudinal scaling for muons (equation 3 and Table 1 in that paper) that yields a variable neutron to muon production rate ratio with changing altitude. Lifton et al. (2014) – EPSL – proposed a new energy-dependent muon scaling but notes that the latitudinal effect of muon scaling is relatively limited. The very recent contribution from Balco (2017) – QG – in turn argues for a new muon parametrisation that lumps fast and stopping muon production pathways in a single term and argues for a geographic scaling depending only on altitude (eq. 7 in that paper). The effect of this new model for determining catchment-wide denudation rates needs to be fully assessed. It has not been implemented in this contribution and it is expected to induce only minor differences that are below the overall uncertainty that we report.*

Page 5 line 34 The absence of incision could mean that there is some addition of floodplain sediment and 10Be concentration to the river bedload by lateral erosion. I guess this would not change your results, but you may discuss this.

*We mentioned this possibility:*

**p.5 l.33:** Lateral channel migration may nevertheless recycle floodplain sediment but the fluxes are overall supposed to be small and the effect of this recycling only marginally increases the 10Be budget of sediments in the floodplain as was suggested for the neighboring Ganga catchment (Lupker et al., 2012).

Page 9 line 7 Please justify why this recent sediment input is unlikely to dominate the sediment budget.

*We added:*

**p.9 l.18:** Some remnant sediments of this event are still present in the channel downstream of the breach (Lang et al., 2013), but over 12 years after this catastrophic flood it is unlikely to they still dominate the sediment budget at the time of sampling since most of this input has likely been evacuated.

Page 10 line 2 Do you consider that sand is mainly in suspension ?

*It depends on the hydrological regime of the river. During high flow, it is likely that a significant fraction of the sand load is transported as suspended sediment. The active bar sediment that we sampled is commonly dominated by sand-sized sediment that is likely deposited after high flow episodes. However, the fact that sand mainly travels as suspended or bedload does not have an impact on the abrasion law we use.*

Page 11 Please justify the range of tested abrasion parameters.

*The parameters tested are based on the parameters that have been obtained in experimental studies for lithology close the one expected in the NBGPm. The number of these studies remains limited and the abrasion parameters difficult to constrain in natural systems but the tested parameters cover a range of 50 to 200 % of the experimentally determined values.*

**p.11 l.22:** The range of abrasion parameters, k, fg, fa and fTCN are estimated from abrasion experiments and grain-size determinations of landslide material (Attal & Lavé, 2006; 2009; Nibourel et al., 2015).

Page 11 line 31-32 I do not understand this sentence.

*The sentence was rephrased:*

**p.11 l.40:** However, our model only takes into account abrasion along the main-stream of the Tsangpo-Brahmaputra and does not take into account the abrasion of landslide material that may have occurred in smaller tributaries before reaching the trunk stream. Our model may therefore slightly overestimate the role of abrasion in generating the observed downstream lag in the denudation signal. Testing the limits of this model and directly comparing it to the data in the eastern syntaxis would require the spatial distribution of landslides and drainage area to be modeled explicitly, which is beyond the scope of this contribution.

Page 11 line 34. I think you should not too much worry about the absence of systematic correlation between grain size and 10Be concentration in this case. This comparison concerns very fine fractions, with a lot of different possible origins. It is possible that the by-products of abrasion that affect the riverine 10Be signal downstream derives from the abrasion of large pebbles and cobbles. Obviously you can not prove this because you do not show 10Be concentration of this fraction, but Carretier et al. (Quaternary Geochronology, 2015) for example, showed a systematic low 10Be concentration in 10 cm pebbles in several catchments along the western Andes, whereas the 10Be concentration of 2 cm gravel is either lower or higher than the sand fraction. In an Himalayan catchment, Puchol et al (2014) showed lower 10Be concentrations in [0.47- 4] cm pebbles. This remains to be proven, but if rapidly eroding areas provide a large proportion of sediment as large pebbles and cobbles, systematically less concentrated in 10Be, then this may be the abrasion of that coarse material that controls the dilution process downstream, and not that of the smaller fraction. If true, knowing the size distribution of material eroded from the hillslopes would be essential to quantitatively interpret the 10Be riverine signal in terms of abrasion rate. Again, it is clear that you cannot provide more information about this debate with your data, but it may be worth to think about it.

*We mentioned the findings by Puchol et al. and Carretier et al. but agree that we cannot go beyond our current interpretation based on the data we present.*

**p.12 l.12:** Our simplified model could actually only be validated or invalidated by comparing [10]Be concentration across a large range of grain-sizes, i.e. between sand size content and boulder size content similarly to what has been done by Puchol et al., (2014) and Carretier et al., (2015b) that found lower 10Be concentrations in pebble sized sediments.

Page 16 line 30 takinG
*Corrected*

Figure 3: localize the Namche Barwa-Gyala Peri massif on that profile and the others.
*The figures have been updated*

Figure 7c: wrong legend.
*Not sure this is wrong?*
sensitivity to changes in the abrasion rate coefficient k;

Figure 10: I do not fully understand what are the two red lines, the red circle and the red zone between the two lines. The caption indicates incorrectly that the x axis is the upstream drainage area whereas it is the downstream distance. "exists" -> exits.
*The caption was corrected and clarified*

[revised manuscript text omitted]

---

## Author Response (AR2)

**Answer to editor for manuscript esurf-2017-18:**
*[10]Be systematics in the Tsangpo-Brahmaputra catchment: the cosmogenic nuclide legacy of the eastern Himalayan syntaxis*

Dear Simon Mudd,

Thanks again for the efficient and thorough handling of the manuscript. We just implemented your recommendations in the last version of the manuscript. This includes i) rephrasing or splitting of awkward formulations, ii) slight rewriting of the abstract and discussion paragraph to clarify our approach, iii) updated figures with larger symbols for better clarity and iv) reference to the work of Lukens et al. (2016) that slipped through the first versions of the manuscript.

I hope you find these reviews

Maarten Lupker (on behalf of all co-authors)

[revised manuscript text omitted]

---

## Author Response (AR3)

Dear Simon Mudd and Josh West,

Thanks for all your inputs, I corrected the spelling and typos that were detected.
Best regards

Maarten